# ReLAM: Learning Anticipation Model for Rewarding Visual Robotic Manipulation

Nan Tang [1 2]   Jing-Cheng Pang [1]   Guan-Lin Li [1]   Chao Qian [1]   Yang Yu [1 2]

## Abstract

Reward design remains a critical bottleneck in visual reinforcement learning (RL) for robotic manipulation. In simulated environments, rewards are conventionally designed based on the distance to a target position. However, such precise positional information is often unavailable in real-world visual settings due to sensory and perceptual limitations. In this study, we propose a method that implicitly infers spatial distances through keypoints extracted from images. Building on this, we introduce Reward Learning with Anticipation Model (ReLAM), a novel framework that automatically generates dense, structured rewards from action-free video demonstrations. ReLAM first learns an anticipation model that serves as a planner and proposes intermediate keypoint-based subgoals on the optimal path to the final goal, creating a structured learning curriculum directly aligned with the task's geometric objectives. Based on the anticipated subgoals, a continuous reward signal is provided to train a low-level, goal-conditioned policy under the hierarchical reinforcement learning (HRL) framework with provable sub-optimality bound. Extensive experiments on complex, long-horizon manipulation tasks show that ReLAM significantly accelerates learning and achieves superior performance compared to SOTA methods.

## 1. Introduction

Reward design stands as one of the most fundamental challenges in reinforcement learning (RL), particularly in the domain of vision-based robotic manipulation (Tian et al.,

2023; Lu et al., 2025; Escontrela et al., 2023; Huang et al., 2024; Pang et al., 2025). In simulated environments, a common and often effective approach is to engineer dense reward signals based on precise geometric information, such as the Euclidean distance between a robot's end-effector and a target position. However, this paradigm faces a critical limitation in real-world applications: exact state information is typically unavailable due to sensory noise, occlusions, and perceptual ambiguities. Consequently, agents must rely on high-dimensional visual observations, making hand-engineered reward design not only labor-intensive but also notoriously challenging. This reward specification bottleneck severely impedes the scalability and adoption of RL in practical robotic settings.

Some prior works overcome this limitation by adopting Learning from Observation (LfO) approaches. A common practice is to employ adversarial frameworks (Torabi et al., 2018; Kostrikov et al., 2019; Xu et al., 2024), where a discriminator that does not take action as input is trained and subsequently used as a reward function. However, when dealing with high-dimensional visual inputs, such methods suffer from significant challenges in terms of training difficulty and stability. In recent years, several works have instead attempted to design visual rewards based on task progress (Zhang et al., 2025a; Zhai et al., 2025; Liu et al., 2025b) or heuristic strategies (Tian et al., 2023; Sontakke et al., 2023; Ma et al., 2023; Escontrela et al., 2023; Huang et al., 2024). These approaches either yield sparse rewards or lack an explicit structured learning process, making them sensitive to distribution shifts or inefficient for long-horizon tasks with extended periods of partial observability and complex dynamics. Thus, there still remains a need for a framework that can automatically synthesize informative, dense reward signals from readily available video demonstrations, while guiding the agent through a structured and geometrically grounded learning curriculum.

This work follows the recent framework of anticipation RL (Yu, 2025) that decomposes the task into short-horizon subtasks. We introduce Reward Learning with Anticipation Model (ReLAM), a novel framework that automatically generates dense and structured rewards from action-free video demonstrations. ReLAM begins by extracting task-

[1]National Key Laboratory for Novel Software Technology, Nanjing University, China & School of Artificial Intelligence, Nanjing University, China [2]Polixir.ai, China. Correspondence to: Yang Yu <yuy@nju.edu.cn>.

*Proceedings of the 43$^{rd}$ International Conference on Machine Learning*, Seoul, South Korea. PMLR 306, 2026. Copyright 2026 by the author(s).

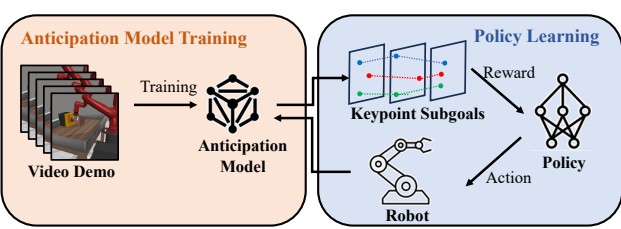

*Figure 1.* An illustration of ReLAM . **Left Part:** The anticipation model is trained from video demonstrations. **Right Part:** The policy learning process with anticipation model.

relevant keypoints as in (Wen et al., 2024) from video demonstrations: we first use the Segment Anything Model (SAM) (Zhang et al., 2024) to isolate objects of interest, then apply a tracking model (Karaev et al., 2024b) to follow pixel-level features across frames. A sparse set of representative points is selected and propagated consistently, forming a trajectory of keypoints that encode object motion. From these, we identify keyframes that signify critical stages of the task, and define the keypoint configurations in those frames as subgoals. Using this curated dataset, ReLAM learns an anticipation model capable of predicting a sequence of intermediate keypoint-based subgoals that lead to the final goal. This model acts as a high-level planner, constructing a structured curriculum aligned with the geometric requirements of the task. The anticipated subgoals then enable the computation of a continuous reward signal based on keypoint distance, which is used to train a low-level, goal-conditioned policy under the hierarchical RL (HRL) framework with provable sub-optimality bound.

Our contributions are as follows: First, we implement the anticipation model (Yu, 2025) by using the distances between learned keypoints, which can provide a meaningful reward signal for RL. Second, we introduce ReLAM, a novel framework that uniquely combines this keypoint-based reward with an anticipative generative model to automatically construct a structured learning curriculum from mere video demonstrations, entirely without action labels. Third, our method bridges the gap between high-level planning and low-level control within an HRL framework, where the anticipation model proposes geometrically meaningful subgoals and the dense, keypoint-derived reward signal reliably guides policy optimization with provable sub-optimality bound. Finally, through extensive empirical validation, we demonstrate that this approach not only significantly accelerates learning but also achieves new state-of-the-art performance on long-horizon tasks, thereby offering a robust and practical pathway toward scalable visual reinforcement learning for robotics.

## 2. Related Work

### 2.1. Robotic Manipulation with Visual Input

Robotic Manipulation with visual input has long been a prominent research topic(Zeng et al., 2018; Kalashnikov et al., 2018; Chi et al., 2023; Liu et al., 2025a; Zhang et al., 2026a; Liu et al., 2026). Traditional approaches rely on supervised learning for behavior cloning, and this paradigm has continued to evolve, giving rise to methods such as Diffusion Policy (Chi et al., 2023; Ze et al., 2024) and VLA-based methods (Kim et al., 2024; Black et al., 2024). However, these approaches require large amounts of data and tend to suffer from substantial compounding errors in long-horizon tasks. In light of these issues, many studies have adopted reinforcement learning to train control policies based on visual input. For example, VPG (Zeng et al., 2018) and QT-Opt (Kalashnikov et al., 2018) apply the vision-based reinforcement learning framework to learn a grasping policy. Recent works (Ren et al., 2025; Lu et al., 2025) have employed RL to diffusion policy or VLA model, demonstrating promising performance in robotic manipulation tasks. Although these image-based reinforcement learning methods show considerable promise, they share a common challenge: the difficulty of reward design. Recently, ATM (Wen et al., 2024) abstracted images into a set of representative keypoints as task representations and employed behavior cloning to train policies, demonstrating strong generalization capability. Motivated by their method, we propose a keypoint-based reward learning approach, which provides an effective solution to the challenge of reward design for robotic manipulation.

### 2.2. Reward Learning from Videos

A common source of reward functions in visual reinforcement learning is the extraction of signals from videos, particularly from expert video demonstrations. Some adversarial imitation learning approaches (Li et al., 2017; Torabi et al., 2018; Rafailov et al., 2021; Kostrikov et al., 2019; Zhang et al., 2025c) employ the output of a discriminator as the reward function; however, such methods often exhibit instability when handling high-dimensional inputs. Benefiting from recent advances in foundation models, a number of works (Tian et al., 2023; Ma et al., 2023; Sontakke et al., 2023) instead use the distance between observations and target images/videos in the representation space as rewards. These approaches usually require pre-given target images, which limits their applicability to open-ended tasks. Some other methods (Escontrela et al., 2023; Huang et al., 2024) indirectly leverage the model's confidence in its generated outputs as a reward signal. Such methods rely entirely on generative models and lack a substantive understanding of the spatial and temporal structures of the task. As a result, they continue to exhibit constraints in un-

seen areas. Besides, some approaches (Zhang et al., 2025a; Zhai et al., 2025; Liu et al., 2025b) directly train the reward model to predict the task progress, which serves as natural feedback to the agent. These progressive indicators work well for in-distribution observations, but might fail in out-of-distribution situations due to the limitation of the expert-only training data. P2R (Shi et al., 2025) introduces the use of distances between pixel-level points as a reward signal. However, it directly uses distance to points in final frame as reward and employs it only for policy evaluation. In contrast, ReLAM effectively extracts structural information across different dimensions of the task from video demonstrations, simplifying the task into point-to-point movements between divided sub-tasks and thereby yielding more generalizable rewards for long-horizon RL.

## 2.3. Hierarchical Reinforcement Learning

Hierarchical reinforcement learning (HRL) aims to improve scalability and efficiency in long-horizon tasks by introducing temporal abstractions. Early frameworks such as Options (Sutton et al., 1999) and MAXQ (Dietterich, 2000) formalize sub-task structures through temporally extended actions and value function decomposition. More recent works focus on goal-conditioned hierarchies and often employ a high-level policy, which can be either a neural network (Nachum et al., 2018; Chane-Sane et al., 2021) or even some foundation models (Pang et al., 2023), to generate subgoals and a low-level policy to execute. It is argued that the high-level policy, which can be called as an anticipation model (Yu, 2025), should identify a waypoint that lies on an optimal shortest path to the final goal to find a global optimal policy. In this work, we will leverage the geometric priors inherent in robotic manipulation tasks to learn an anticipation model capable of continuously generating subgoals and train policy under the HRL framework.

## 3. Method

This section presents the method ReLAM which automatically provides reward by learning from video demonstrations $\mathcal{D} = \{V_i = (I_1^i, I_2^i, \cdots, I_{T_i}^i)\}_{i=1,\cdots,N}$. We divide our approach into two stages. The first stage is to learn an anticipation model which takes in the current task state and desired final goal as input to produce an easy-to-reach intermediate keypoint-based subgoal. At the second stage, with the assistance of the anticipation model, a dense reward function is designed to train a low-level, goal-conditioned policy. We will elaborate on these two stages below.

## 3.1. Anticipation Model Learning with Keypoints

This part introduces how we learn an anticipation model from video demonstrations. Instead of training the anticipation model to generate images, we simplify it into a key-

point generation model like ATM (Wen et al., 2024). A good selection of keypoints can be a highly abstract and effective representation of the task, and will reduce the difficulty of generating subgoals simultaneously. In the following section, the learning procedure of keypoint-based anticipation model will be presented by answering three questions: (1) How to select representative keypoints? (2) How to determine an appropriate subgoal for anticipation model to generate? (3) How to train the anticipation model?

### 3.1.1. SUBGOAL DATASET GENERATION

**Keypoint Selection** For the first question, i.e., to select the representative keypoint in one image, it is important to first pick out the key objects. ATM (Wen et al., 2024) samples pixels averagely in one image, which might make too many points chosen, leaving these points unrepresentative. In ReLAM, we propose a new sampling strategy to elect the representative keypoints. First, for each video demonstration, we extract its first frame and apply a grounded SAM model (Zhang et al., 2024) to obtain task-relevant segmentations. Next, for the pixels corresponding to each key object in the image, we employ a track model (Karaev et al., 2024a) to follow their motion trajectories throughout the entire video demonstration. Specifically, each trajectory records the two-dimensional coordinates of the pixel within the image coordinate system across all frames of the demonstration. Among all pixels corresponding to key objects in the image, we identify those that are truly relevant to the task by applying a predefined threshold to remove pixels whose motion range across the video is negligible. After filtering out pixels with small displacements, we further select the final keypoints using Farthest Point Sampling (FPS) (Eldar et al., 1997). The entire procedure for keypoint selection can be summarized by the following formulation:

$$\mathcal{P} = \mathbf{FPS}\left(\{p = (x, y) \in SAM(I_0) : \max_{0 \leq t, t' \leq T} d_{t,t'} \geq \Theta\}\right) \tag{1}$$

For the equation above, $I_0$ represents the initial frame of the video, $SAM$ denotes the segmentation model that picks out the task-relevant pixels, $d_{t,t'} = (x_t - x_{t'})^2 + (y_t - y_{t'})^2$ with $(x_t, y_t)$ means to which position the point $(x, y)$ in $I_o$ will move at time $t$ in the video, $\Theta$ is a predefined threshold and $\mathbf{FPS}$ denotes the Farthest Point Sampling technique. In most robotic manipulation tasks, these points serve as a high-level abstraction of the task state. By tracking the motion of these keypoints, one can infer the location and posture of the robotic arm, as well as whether it has performed the intended action on the object.

**Keyframe Selection** Filtering task-relevant keypoints in the image simplifies and condenses the spatial structure of the task. For a video sequence, however, the temporal dimension is equally important, as it reveals the underlying

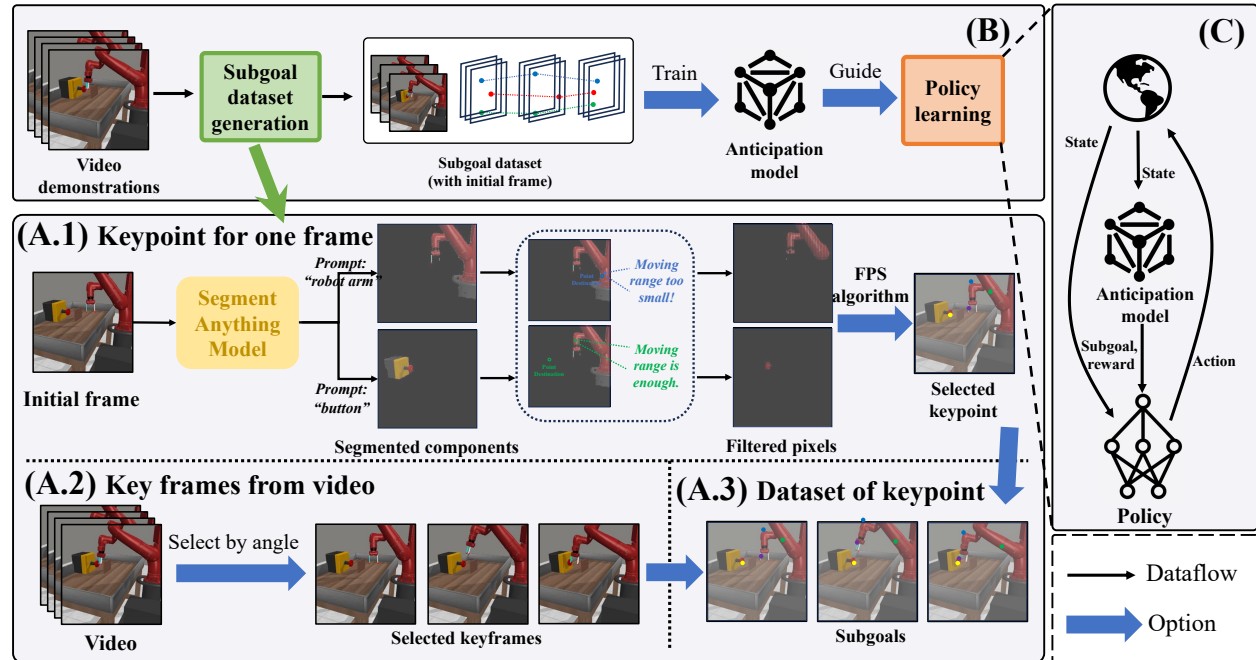

*Figure 2.* Overall training framework ReLAM method. (A) ReLAM first picks out representative keypoints in the initial frame of the video and then selects keyframes throughout the video, turning the position of keypoints in these frames into subgoals. (B) Training the anticipation model based on the generated subgoal dataset. (C) Training policy with point-based reward with subgoals generated from anticipation model.

logic and patterns of the robot arm's motion.

In ReLAM, keyframes are picked out from the video demonstration every certain interval. Specifically, we pre-define a minimum step size $m$ and a maximum step size $M$. We then track the movements of the keypoints and, within the step range $[m, M]$, identify the timestep at which the change of keypoint motion is most pronounced. Specifically, we determine them based on the angle between the displacement vector of the current timestep and that of the previous timestep: usually a keyframe lies at the boundary between two consecutive simple motions for the robot where the angle becomes significantly larger, in which case the frame is regarded as a keyframe. This keyframe selection process can be formalized as follows:

$$t_j = \arg \min_{t \in [t_{j-1}+m,\, t_{j-1}+M]} \sum_{k=1}^{K} \frac{\langle p_t^k - p_{t-1}^k, p_{t+1}^k - p_t^k \rangle}{\|p_t^k - p_{t-1}^k\| \|p_{t+1}^k - p_t^k\|} \quad (2)$$

where $t_j$ is the timestep for $j$-th keyframe, $p_t^k$ denotes the coordinate of the $k$-th keypoint at timestep $t$ and $\langle \cdot, \cdot \rangle$ is the inner product operation. For each video demo, we take the keypoints extracted from the initial frame using Eq. (1) and track their coordinates across the video keyframes obtained via Eq. (2). In this way, we construct the keypoint dataset below, with $p_i^k$ being the position of $k$-th keypoint at keyframe $j$ for demo $i$, and $x_{i,j}^k, y_{i,j}^k$ being its correspond-

ing coordinate.

$$\mathcal{K} = \bigcup_{i=1}^{N} \mathcal{K}_i = \bigcup_{i=1}^{N} \{p_{i,j}^k = (x_{i,j}^k, y_{i,j}^k)\}$$

### 3.1.2. ANTICIPATION MODEL LEARNING

The dataset $\mathcal{K}$ can be regarded as a collection of subgoal sequences composed of keypoint coordinates. Therefore, we employ an autoregressive model as the anticipation model to generate these subgoals sequentially. The anticipation model takes the initial visual observation $I_0$ of the task as input and performs two steps: (1) it identifies the keypoints within $I_0$ and records their coordinates $P_0$; (2) based on $I_0$ and $P_0$, it autoregressively predicts the coordinates of these keypoints in the subsequent keyframes.

For image inputs, previous research (Zhou et al., 2024) have shown that directly leveraging representations from pretrained vision models often endows the model with stronger spatial understanding, thereby enhancing its generalization capability. Motivated by this observation, we also adopt a frozen DINOv2 (Oquab et al., 2023) model to extract image embeddings. The visual input for anticipation model here is one RGB image of the initial frame with size $256 \times 256$. After being processed by the DINOv2 model, the image is divided into $16 \times 16$ patch embeddings. These patch embeddings are then concatenated with the tokens formed by the coordinates of keypoints from historical

keyframes and fed into the model. The coordinates of the keypoints are first normalized and then mapped through a Multilayer Perceptron (MLP) into the embedding dimension. In the case of the first step, where no historical keypoints exist, we instead use a fixed special token to indicate that the model should predict the keypoint positions based on the given images. After being fed into the model, the image and point embeddings will pass through 12 layers of causal transformer blocks. Subsequently, the tokens corresponding to the points are processed by a MLP to predict the point coordinates in a residual form. These predicted coordinates are then compared with the ground-truth coordinates using Mean Squared Error loss under a teacher-forcing scheme to train the anticipation model. The structure of the anticipation model is displayed in Fig. 3.

## 3.2. Policy Learning with Point-based Reward

We train our policy under the hierarchical reinforcement learning framework. At the beginning of each episode, the initial image is fed into the anticipation model trained in the previous section. The model first predicts the keypoints' location $P_0$, and then autoregressively generates a sequence of subgoals (i.e., keypoint) $P_1, \cdots, P_k$. Our objective is to design a reward function that encourages $P_0$ to sequentially move towards $P_1, \cdots, P_k$, thereby enabling the robot arm to successfully complete the task. Based on the assumption that motion between keyframes is simple, the reward can be directly defined using the Euclidean distance in the pixel coordinate system. Formally, we define the movement of a subgoal from $P_{j-1}$ to $P_j$ as the $j$-th stage. For this stage, the $L_2$ distance between the current position of the keypoint and the subgoal $P_j$ is computed as $l_j$. Next, a monotonic function is employed to transform it into dense reward $r_{\text{dense}}$. We find that a piecewise linear function performs best and the results can be seen in Fig. 6(b). We assume that when the distance between a keypoint and the subgoal is smaller than a predefined threshold $\theta_j$, the robot is considered to have successfully achieved the subgoal of stage $j$. At this point, the process transitions to stage $j + 1$, with the subgoal updated to the $(j + 1)$-th target position. Upon completing each subgoal, the robot receives an additional stage-success reward, and upon accomplishing the entire task, it is granted a final success reward. Consequently, the overall reward can be expressed as follows:

$$r = r_{\text{dense}} + r_{\text{success}} + I(l_j \leq \theta_j) \qquad (3)$$

We find that when trained with this kind of reward, the policy is able to find a near-shortest path to complete the task. We provide a brief mathematical proof to show this near-optimality in App. D.

## 4. Experiments

In this section, we conduct extensive experiments to evaluate our proposed method in robotic manipulation tasks. We first introduce the experiment setup.

### 4.1. Experiment Setup

**Evaluation environments.** We conduct experiments on two robotic manipulation environments: Meta-World (Yu et al., 2019) and ManiSkill (Gu et al., 2023), as shown in Fig. 4. **(1) Meta-World:** This environment requires the agent to control a Sawyer robotic arm with 7 degrees of freedom (DoF) and a parallel finger gripper. Meta-World offers a suite of 50 distinct manipulation tasks, covering a wide array of scenarios, such as interactions with drawers, buttons, doors and plates. **(2) ManiSkill:** ManiSkill is a powerful unified framework for robot simulation and training powered by SAPIEN. Here we focus on table-top manipulation tasks, which involve a Panda robotic arm by Franka Emika with 7 DoF and a parallel finger gripper. These tasks are primarily focused on block manipulation tasks. **(3) ALOHA**: We include a real-world ALOHA robot to fold a towel on the table. The task is decomposed into three sequential stages: (a) approaching the right towel and pulling it to the middle; (b) folding the towel in half; and (c) performing a second fold. We mainly use Drawer Open, Door Open, Button Press Wall, Coffee Button, Faucet Open, Peg Insert Side, Plate Slide from Meta-World, and Push Cube, Pick Cube, Push T from ManiSkill for evaluation. The observations on all tasks are images with $256 \times 256$ pixels, which are captured by the fixed-position third-person camera. We run online RL and offline RL for both these environments.

**Dataset for training.** Video demonstration dataset contains 100 trajectories collected by motion planning for each task in both Meta-World and ManiSkill. This video dataset is action-free and used to generate keypoint subgoal dataset for the training of anticipation model. Besides the video demo dataset, we also have an offline control dataset which contains action for offline reinforcement learning setting with 200 trajectories gathered for each task. Among these trajectories, 100 of them are expert demonstrations and another 100 are obtained by adding random noise to expert action.

**Implementation details.** For online RL, We build upon the well-established open-source reinforcement learning library Stable Baselines3 (Raffin et al., 2021), utilizing its PPO implementation. For offline RL setting, we utilize OfflineRL-kit (Sun, 2023), a well-verified offline RL codebase. Specifically, we use Implicit Q-Learning (Kostrikov et al., 2022), an offline reinforcement learning algorithm that avoids explicit policy constraints by learning value functions implicitly and extracting a policy

through advantage-weighted regression. We use 16 CPU cores (AMD EPYC 9654 @ 2.4GHz) and 1 GPU (NVIDIA GeForce RTX 4090) for our experiments. More implementation details can be found in Appendix B.

## 4.2. Main Results

**Baselines for comparison** We choose the following representative approaches which learn a reward from videos for comparison. (1) **DACfO** is an adversarial imitation learning method which combines the idea of DAC (Kostrikov et al., 2019) and GAIfO (Torabi et al., 2018), where we modify the discriminator's input like GAIfO to consist of the current observation $o$ and the next observation $o'$, enabling it to handle action-free demonstration datasets. For offline data, we first run DACfO online and save the last 10 checkpoints of the discriminator. Then we use them to label the offline dataset with ensemble technique. (2) **Diffusion Reward (DR)** (Chi et al., 2023) trains a diffusion model with the video demo data and learn a policy by computing the conditional entropy of the diffusion model as reward. (3) **ReWiND** (Zhang et al., 2025a) trains a progressive reward model by randomly rewinding the expert videos. (4) **Image Subgoal (IS)** integrates our method with the idea of VP2 (Tian et al., 2023) by employing a flow matching model to autoregressively generate subgoals from the initial image, and then uses the cosine similarity between the current visual observation and the target image in the representation space of DINOv2 (Oquab et al., 2023) as the reward to train a goal-conditioned policy with image subgoal. (5) **Oracle** replaces the generated image subgoals in IS baseline with the ground-truth ones for each episode, with all other components unchanged.

**Results for Online Reinforcement Learning.** Tab. 1 shows the success rate of different reward learning methods for online reinforcement learning results in both Meta-World and ManiSkill environments. In general, our proposed method ReLAM outperforms the baselines for all three environments. It can be observed that on these tasks, ReLAM rapidly achieves very high success rate. In contrast, other baseline methods either fail to reach such high success rates or require significantly more interaction steps. We evaluate for five fixed seeds ($0 - 4$), and it is worth noting that for some seeds, the Diffusion Reward approach completely fails to learn. This occurs because their method relies on an auxiliary RND reward to encourage exploration, which does not necessarily provide a correct exploration signal and instead results in highly stochastic exploration. Under such circumstances, certain seeds may never encounter the correct trajectory, ultimately preventing successful learning. In contrast, our method incorporates both the current and target coordinates of keypoints as part of the policy input, inherently providing the policy with implicit guidance. Moreover, the distance-based reward en-

ables the policy to gradually recognize that approaching the target yields higher returns, thereby steering exploration toward meaningful regions of the state space and allowing the agent to acquire the task more efficiently. For ReWiND, the progress reward is trained directly on expert demonstrations. Although the rewind technique can augment the training data to some extent, the resulting data distribution remains narrow, making the progress reward unreliable on failed exploration trajectories. In contrast, ReLAM incorporates a spatial-relation prior, which substantially alleviates the difficulty of reward prediction outside the expert data distribution, thereby leading to superior policy performance. For DACfO, the curve exhibits substantial fluctuations, indicating that the training process is indeed unstable. For Oracle, since it leverages ground-truth subgoal images, we can see that learning proceeds relatively quickly and ultimately achieves a high success rate. In contrast, when we replace the subgoals with results generated by the flow matching model, the performance, as shown by IS, drops significantly. The generated images often contain noise and local blurriness, making it difficult to establish a consistent similarity threshold in the representation space. For example, while a threshold of $0.95$ may be appropriate for the first generated result, the second might require $0.9$, and this threshold can vary further depending on the initial state of the task. Due to this inconsistency, the IS method achieves very low success rates, with only a small fraction of well-generated cases for the policy to learn.

**Results for Offline Reinforcement Learning.** Tab. 2 displays the success rate of different approaches for offline reinforcement learning results in ManiSkill environments. On Pick Cube and Push Cube tasks, our method surpasses all baseline approaches except Oracle which cheats with the ground-truth image subgoal. It can be observed that ReLAM achieves performance comparable to Oracle, whereas IS shows a clear performance drop compared to Oracle. This indicates that even without access to privileged information, by leveraging abstract keypoints as targets, ReLAM not only reduces the generation difficulty for the anticipation model but also effectively captures structural information of the task to guide the policy. The reason why IS achieves much higher success rates than Meta-World is that: (1) The offline control dataset contains expert action label, reducing the need for exploration; (2) The sequence of subgoals for ManiSkill is shorter, leading to less compounding error of the generation results anticipation model. The presence of successful trajectories in the dataset also helps explain why ReWiND exhibits improved performance in some cases under online settings. The performance of Diffusion Reward and DACfO seems similar for some tasks, as both methods share essentially the same underlying principle: rewards are higher near the expert distribution and lower when further away from it. Since a

|  | Diffusion Reward | ReWiND | DACfO | IS | Oracle | ReLAM |
|---|---|---|---|---|---|---|
| *Meta-World Environments* | | | | | | |
| Drawer Open | 80.0 | 66.0 | 71.3 | 4.0 | 93.3 | 100.0 |
| Door Open | 100.0 | 84.0 | 70.0 | 24.7 | 100.0 | 100.0 |
| Button Press Wall | 60.0 | 65.3 | 47.8 | 12.7 | 76.7 | 75.8 |
| Coffee Button | 100.0 | 100.0 | 88.0 | 96.7 | 100.0 | 100.0 |
| Faucet Open | 80.0 | 86.7 | 40.0 | 16.7 | 96.0 | 94.7 |
| Plate Slide | 80.0 | 100.0 | 41.3 | 6.0 | 100.0 | 100.0 |
| Peg Insert Side | 38.0 | 62.0 | 32.3 | 13.3 | 81.3 | 66.0 |
| *ManiSkill Environments* | | | | | | |
| Push Cube | 60.0 | 40.0 | 46.7 | 23.3 | 100.0 | 84.0 |
| Pick Cube | 42.7 | 30.0 | 19.3 | 4.7 | 82.0 | 56.7 |
| Push T | 31.3 | 29.7 | 7.3 | 0.0 | 42.0 | 38.7 |

*Table 1.* Mean success rate of the last evaluation for each method on training tasks with online RL.

large portion of the offline decision-making data is near-expert, the rewards assigned by both methods are generally high, which explains why they ultimately achieve comparable results. However, their performances still fall short of ReLAM. We think this is because our approach can recognize and reward trajectories that deviate slightly from the expert distribution yet still move effectively toward the goal. This is enabled by our distance-based reward design, which provides the policy with a strong guidance signal and fosters a deeper understanding of the task's spatiotemporal structure. In contrast, Diffusion Reward may assign lower returns to such trajectories due to its higher entropy, leading to less efficient learning.

**Real-world Evaluation.** We apply an IQL-like offline reinforcement-learning algorithm on the human-collected real-world data of the ALOHA robot based upon the reward annotated with ReLAM. The details are referred to Sec. B.5. Tab. 3 presents the experiment results of each method evaluated for 20 trajectories with random initialization of the color of towel and its position. We also consider two baselines for comparison: (1) Vanilla BC which performs behavior cloning on the collected successful trajectories with $\pi_0$; (2) ReWiND which denotes the collected trajectories using its learnt reward model and perform policy learning with IQL. The results show that the proposed method does make a good reward for the policy learning. IQL with either ReWiND or ReLAM improves the performance of the Vanilla BC policy in all three stages. Moreover, ReLAM achieves better policy performance compared to ReWiND, indicating that our approach provides more effective guidance for policy learning through the targets predicted by the anticipation model and the spatial-relation-based reward.

**Learning from Human Videos** We collected 40 human demonstration videos of drawer-opening behaviors analogous to the Drawer Open task in Meta-World, and extracted

the corresponding keypoint subgoal dataset. Starting from this dataset, we found that incorporating only a very small amount of target-domain keypoint data, i.e., keypoint annotations extracted from a few expert demonstrations in Meta-World Drawer Open, is already sufficient to produce an effective reward signal. Fig. 6(d) reports the success rates on Meta-World Drawer Open when training follows the ReLAM pipeline with either 40 human videos plus 1 Meta-World expert video, or 40 human videos plus 5 Meta-World expert videos. These results demonstrate that ReLAM exhibits promising few-shot transfer capability.

**Some other results.** We make a visualization of the rewards for different methods in Fig. 5, which verifies our analysis above. We sample 20 trajectory segments for Pick Cube task and label them with four types of rewards: environment rewards, ReLAM, DR, and DACfO. We then projected the labeled trajectories into a two-dimensional space using t-SNE, where each trajectory corresponds to a single point. Since ReLAM assigns rewards based on keypoint distances and the environment reward is based on distances in the world coordinate system, these two rewards are relatively close. Moreover, as mentioned above, both Diffusion Reward and DACfO assign higher rewards to regions closer to the expert distribution, which explains why their projections are not far away from each other. More results are displayed in App. E.

### 4.3. Ablation Study

**Effect of the point number.** We study whether the number of points selected for the task will affect the performance of the policy. We sample 4, 8 and 12 points for Drawer Open task and the result is shown in Fig. 6(a). The best performance is achieved when sampling four keypoints, followed by eight keypoints, while twelve keypoints yield the weakest results. This outcome can be explained by the truth that for Drawer Open task, sampling four keypoints—three on

|  | Diffusion Reward | ReWiND | DACfO | IS | Oracle | ReLAM |
|---|---|---|---|---|---|---|
| *Meta-World Environments* | | | | | | |
| Drawer Open | 96.0 | 87.3 | 58.7 | 6.0 | 98.0 | 100.0 |
| Door Open | 72.7 | 88.7 | 78.0 | 53.3 | 100.0 | 100.0 |
| Button Press Wall | 96.0 | 98.0 | 94.7 | 80.7 | 100.0 | 98.0 |
| Coffee Button | 100.0 | 96.0 | 98.0 | 81.3 | 100.0 | 100.0 |
| Faucet Open | 26.7 | 50.7 | 20.0 | 16.0 | 79.3 | 61.3 |
| Plate Slide | 80.7 | 82.7 | 66.0 | 54.0 | 100.0 | 93.3 |
| Peg Insert Side | 41.3 | 55.3 | 48.7 | 17.3 | 94.0 | 83.3 |
| *ManiSkill Environments* | | | | | | |
| Push Cube | 69.3 | 82.0 | 76.7 | 46.7 | 92.7 | 89.3 |
| Pick Cube | 68.0 | 84.0 | 78.7 | 60.7 | 90.7 | 88.0 |
| Push T | 41.3 | 46.7 | 39.3 | 24.7 | 63.3 | 54.7 |

*Table 2.* Mean success rate of the last evaluation for each method on training tasks with offline RL.

| Method | Vanilla BC | ReWiND | ReLAM |
|---|---|---|---|
| Stage 1 | 75% | 85% | 90% |
| Stage 2 | 45% | 55% | 65% |
| Stage 3 | 20% | 30% | 35% |

*Table 3.* Results on real-world ALOHA robot.

| Method | Vanilla BC | ReLAM |
|---|---|---|
| Peg Insert Side | 39.3% | 66.0% |
| Pick Cube | 28.0% | 56.7% |

*Table 5.* Results of pure BC on two tasks.

| Azimuth | 31° | 34° | 37° | 40° | 43° |
|---|---|---|---|---|---|
| Prediction error | 2.6 | 7.4 | 12.7 | 28.8 | 67.0 |

*Table 4.* Results of the prediction error under different views.

the robotic arm and one on the drawer—is sufficient to provide an adequate representation of the task state, thereby enabling rapid policy learning. In contrast, with eight or twelve keypoints, the prediction difficulty for the anticipation model increases. Moreover, requiring the policy to simultaneously drive all keypoints toward their respective targets imposes stricter constraints on its actions, effectively forcing the behavior to closely mimic the demonstrations. In this regard, four points can provide a sufficiently broad criterion while allowing more flexible exploration.

**Effect of the reward function.** We evaluate the impact of different types of reward functions conditioned on the keypoint distance have on the performance of reinforcement learning. We introduce three other kinds of reward functions: (1) pure linear function; (2) exponential function; (3) logistic function. All these functions are designed to have the same range when the point distance is between $[0, 30]$. The performance is displayed in Fig. 6(b). We found that the piecewise linear function took the lead and linear function achieved relatively good performance, while the exponential and logarithmic functions performed worse. We think this is because the slopes of the latter two functions vary continuously, making it difficult for the policy to adapt; while the slope of the linear function remains constant, resulting in insufficient encouragement for the policy as it approaches the target. The piecewise lin-

ear function, however, strikes a balance between the two: it provides sufficient incentives for the policy to reach the goal while maintaining a certain degree of stability.

**Generated subgoal versus ground-truth subgoal.** To evaluate the accuracy of the anticipation model in generating subgoals, we compare its predictions with ground-truth subgoals. Specifically, at each environment initialization, we provide the subsequent sequence of ground-truth keypoint subgoals and train a goal-conditioned RL policy based on these targets like ReLAM. Fig. 6(c) presents a comparison between the performance of policies trained with AR-generated subgoals and those trained with ground-truth subgoals. The performance gap between the two is relatively small, indicating that the anticipation model produces sufficiently accurate subgoals. In contrast, when the subgoals are represented as images rather than points, the performance of Image Subgoal drops significantly compared to the Oracle baseline. This result further confirms that point-based representations substantially reduce the difficulty of the generation problem and produce effective rewards to guide the policy.

**Robustness under different viewpoints.** To study robustness to viewpoint variation, we sampled 10 camera viewpoints with azimuth angles randomly chosen between $10°$ and $30°$ on Drawer Open task. For each viewpoint, we collected 10 expert demonstration videos and trained the anticipation model on these videos. We then evaluated policy learning in RL environments with previously unseen viewpoints, using azimuth angles of $31°$, $34°$, $37°$, $40°$, and $43°$. For each of these unseen viewpoints, we collected a small set of ground-truth demonstration videos and extracted a

ground-truth keypoint subgoal dataset. We then used the anticipation model to predict keypoint subgoals from the initial frame of each demonstration and measured the average of the L2 distance between the predicted subgoals and the ground-truth subgoals, as reported in Tab. 4. As the test viewpoint gradually deviates from the viewpoints seen during training, the prediction error of the anticipation model also increases. Moreover, once the viewpoint shift exceeds roughly $10°$, the error increases sharply. This suggests that ReLAM exhibits a certain degree of robustness to viewpoint variation. We also performed RL experiments at the same five unseen viewpoints using the trained anticipation model. As shown in Fig. 6(e), the policy success rate drops substantially at an azimuth angle of $43°$, which is consistent with the trend observed in the above table.

**Robustness under mosaic.** To evaluate the robustness of ReLAM under visual distractions, we introduced mosaic occlusions to the Meta-World Drawer Open task. We considered two types of mosaics, as shown in Fig. 13. In the first setting, the mosaic occludes most of the drawer handle region. In the second setting, the mosaic covers the rear part of the drawer while leaving the handle visible. In both cases, the mosaic size was fixed to $40 \times 40$ pixels. For the expert demonstration videos, we applied each type of mosaic separately and trained the anticipation model following the standard ReLAM pipeline. During RL procedure, the corresponding mosaic was also applied to the environment observations used for policy training. The results, shown in Fig. 6(f), indicate that the second mosaic has almost no impact on policy performance, as the occluded region does not contain the drawer handle. In contrast, under the first mosaic, the handle is fully occluded in some frames, which may cause CoTracker to fail in tracking the relevant keypoints. These results suggest that ReLAM remains effective as long as the task-relevant object parts, such as the drawer handle, are not completely occluded.

**Robustness under tracking noise.** We investigated the effect of tracking errors by adding Gaussian noise with different magnitudes to the coordinates of the points tracked by CoTracker during RL interaction on the Drawer Open task. As shown in Fig. 6(g), larger tracking noise leads to a greater degradation in policy performance. Meanwhile, ReLAM consistently uses the keypoints from the initial frame as the reference for CoTracker, which prevents tracking errors from accumulating over time. This property helps explain why ReLAM remains relatively robust under moderate tracking noise.

**Comparison with BC.** We conducted pure behavior cloning (BC) experiments on Peg Insert Side and Pick Cube tasks using 100 expert trajectories with action labels. For a fair comparison, the BC policy also adopted a CNN backbone and was trained for 200,000 update steps with a batch

size of 128. This setup ensures that BC and the PPO-trained ReLAM policy observe a comparable number of state-action pairs, both approximately $2 \times 10^7$. As shown in Tab. 5, even with a chunk size of 8 used to mitigate the compounding error (Xu et al., 2020), pure BC still underperforms ReLAM trained with only a small amount of expert BC data, namely 10 expert trajectories. We conjecture that this is because the BC loss used during the training of the ReLAM policy provides a good initialization for RL and also serves as a useful regularizer during optimization, while the reward generated by ReLAM continuously guides the policy toward improved task performance.

## 5. Conclusion

This study explores the reward design problem for robotic manipulation. We propose a novel approach, ReLAM, which first learns an anticipation model as a planner proposing intermediate keypoint-based subgoals and then trains a goal-conditioned policy with the distance of keypoints as reward signal. Extensive experiments demonstrate that ReLAM is capable of being applied to a variety of robotic platforms, enabling a robust and practical pathway towards scalable RL for robotic manipulation. Despite the promising results, there are still limitations. One limitation of our work is reliance on the viewpoint shown in the ablation study. Our anticipation model is trained with video demos from a single camera with the assistance of a track model. If the viewpoint undergoes a dramatic change, the model will struggle to generate the desired target. Moreover, significant occlusions can prevent the track model from accurately following the keypoints. A potential solution is to use observation from multi views and merge them into the point cloud, which is more robust to viewpoint disturbance and occlusion. Besides, the experiment scale is limited, in terms of the dataset scale and model size. In future works, we hope to scale up the framework to solve more challenging tasks. For instance, employing a pre-trained VLM or leverageing the knowledge from LLM (Wang et al., 2025) for the anticipation model, and training with more data like Open X-Embodiment (Collaboration, 2023). Last but not least, we hope to apply ReLAM not only to simulator or real world interaction, but also to a world model (Pang et al., 2024; Liu et al., 2026; Zhang et al., 2026b) to assign guidance to generated trajectories. We believe these interesting directions are worth further exploration for developing smarter robots with the support of more general-purpose reward.

## Acknowledgements

This work was supported by the National Natural Science Foundation of China (62495090, 62495093), the Natural Science Foundation of Jiangsu (BK20243039), the "111

Center" (No. B26023), the Fundamental and Interdisciplinary Disciplines Breakthrough Plan of the Ministry of Education of China (No. JYB2025XDXM118). The authors thank anonymous reviewers for their helpful discussions and suggestions for improving the article.

## Impact Statement

This paper presents work whose goal is to advance the field of Machine Learning. There are many potential societal consequences of our work, none of which we feel must be specifically highlighted here.

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

# Appendix

## A. The Structure of the Anticipation Model

We visualize the structure of the anticipation model in Fig. 3.

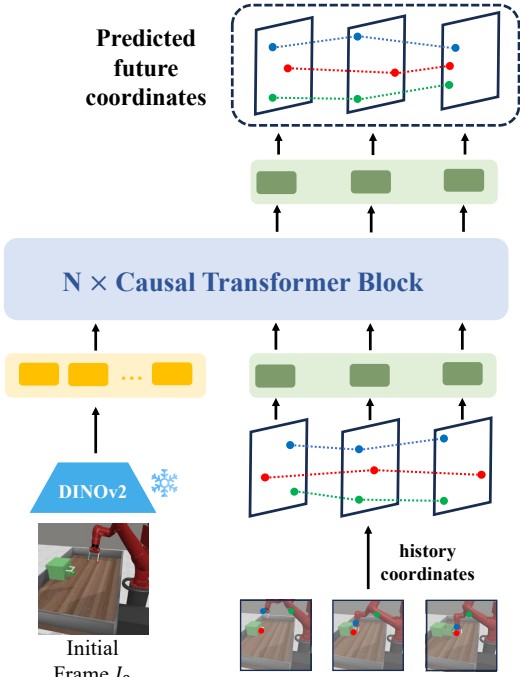

*Figure 3.* The structure of the anticipation model for subgoal generation.

## B. More Implementation Details & Experiment Setup

### B.1. More Details for Learning Anticipation Model

After obtaining the segmentation of task-relevant objects in the keypoint selection stage, we perform an additional filtering step: a point is retained only if it, along with all the pixels within an $L \times L$ square centered on it, lies inside the segmentation. This is because the SAM model sometimes includes extra pixels from the background or other irrelevant objects, which usually appear as isolated rather than contiguous regions. The above operation effectively filters out such points. To prevent severe occlusion of task-relevant objects, we have two viewpoints for Meta-World tasks, with the azimuth being $12.5$ or $167.5$. For Drawer Open, Button Press Wall, Faucet Open, Coffee Button and Plate Slide, the azimuth of the camera is set to be $12.5$, and for Peg Insert Side and Door Open, the azimuth is set to be $167.5$.

As for the track model, It is important to note that it is capable of following points that initially appear within the image boundaries but later move outside the frame. In other words, assuming the image size is $H \times W$, the coordinates of a point $(x_t, y_t)$ may take values such as $x_t < 0$ or $y_t > W$. This property further ensures the model's robustness in tracking keypoints and extends its effective tracking range.

### B.2. More Details For Policy Learning

As we say, various monotonic functions can be used to form a keypoint distance-based reward, such as exponential, logarithmic, or linear functions. But We find that a piecewise linear function performs best. We provide the specific form below, where we make sure all these functions have the same range when the distance is in $[0, 30]$. For the piecewise linear

function, we use:

$$r(l) = \begin{cases} 0, & l \le 6 \\ -0.05 \cdot l + 0.3, & 6 < l \le 14 \\ -0.025 \cdot l - 0.05, & 14 < l \le 30 \\ -0.01 \cdot l - 0.5, & l > 30 \end{cases} \tag{4}$$

For linear function, we use:

$$r(l) = \begin{cases} 0, & l \le 6 \\ -(l-6)/30, & l > 6 \end{cases} \tag{5}$$

For exponential function, we have:

$$r(l) = \begin{cases} 0, & l \le 6 \\ -0.8 \cdot (1 - e^{6-l})/(1 - e^{-24}), & l > 6 \end{cases} \tag{6}$$

For logistic function, we use:

$$r(l) = \begin{cases} 0, & l \le 6 \\ -0.8 \cdot (\log l - \log 6)/\log 5, & l > 6 \end{cases} \tag{7}$$

During inference, we do as the training stage: first generate the subgoal sequences with anticipation model, then instruct the policy to complete them one by one.

## B.3. More Details for PPO

We make some modifications based on the source code of PPO in Stable Baselines3. To make the collecting process compatible with hierarchical reinforcement learning framework, we set $terminal = True$ once a subgoal is achieved or a whole episode ends. This operation segments the whole trajectory by subgoal, which makes GAE computation done separately for each low-level policy. Besides, we add reward scaling technique to make learning faster and more stable. For policy and critic network, the input consists of the current RGB observation $I$, the current coordinate of the keypoints $p$, and the target coordinate of the keypoints $p'$ predicted by anticipation model. For image $I$, we utilize a three-layer CNN network as encoder. For the points $p$ and $p'$, we flatten and feed them into a MLP to get point representation. The image and point features are concatenated together and sent into another MLP to get the final output.

In *Peg Insert Side* and *Pick Cube* tasks, it is extremely challenging for the training-from-scratch policy to learn with its own exploration. To facilitate efficient learning, we incorporate a small set of $N = 10$ expert demonstrations into the online training process for our method and all evaluated baselines. Specifically, the standard RL objective is augmented with a Behavioral Cloning (BC) auxiliary loss:

$$\mathcal{L}_{total} = \mathcal{L}_{RL} + \lambda \mathcal{L}_{BC} \tag{8}$$

where $\mathcal{L}_{BC} = -\mathbb{E}_{(s,a) \sim \mathcal{D}_{exp}}[\log \pi_\theta(a|s)]$ represents the imitation loss over the expert dataset $\mathcal{D}_{exp}$, and $\lambda$ is a weighting coefficient that balances exploration and expert guidance. For other tasks, no expert demonstrations are provided during online training of the policy.

## B.4. More Details for Offline RL

For offline RL with ReLAM, we first generate the desired subgoals using anticipation model for each trajectory. Then a similar way to online RL is employed to label the reward with distance like Eq. 3, and we will proceed to the next subgoal if the distance is within the threshold. For IS and Oracle baseline, we assign rewards in the same way to ReLAM. For DACfO, we first run online DAC in the task environment and save the last 10 checkpoints of the discriminator. After online training, rewards are given by the average of the output from these 10 discriminators, which is found to be more generalizable than using only one discrimator. (Luo et al., 2022).

**B.5. More details for Real-world Evaluation.**

We designed a towel-folding task using the ALOHA bimanual robotic platform, which features a multi-camera system comprising a head-mounted camera and two wrist-mounted cameras. Five towels of different colors (pink, green, blue, white, and yellow) were used. In each session, two towels were sampled and positioned on the left and right sides of the workspace. The robot received linguistic instructions to manipulate a target towel, e.g., "*fold the pink towel*". The task setting is shown in Fig. 4.

Under this setup, a total of 40 unique task combinations were defined. We collected two trajectories per combination, resulting in a dataset of 40 successful and 40 failed trajectories. For ReLAM, we first extracted a keypoint dataset from the head-camera views of the successful demonstrations to train an anticipation model. This model was then used to assign rewards to all 80 trajectories, enabling the training of a goal-conditioned policy via Implicit Q-Learning (IQL) (Kostrikov et al., 2022). For the **Vanilla BC** baseline, we performed behavior cloning exclusively on the 40 successful trajectories. For **ReWiND**, consistent with our approach, we trained a reward model on successful demos and applied IQL over the full 80-trajectory set. All models were built upon the $\pi_0$ (Black et al., 2024) backbone and fine-tuned using Low-Rank Adaptation (LoRA) with standard hyperparameters.

In ReLAM, long-horizon tasks and restricted camera fields-of-view often lead to partial observability, where not all task-relevant objects are visible throughout the execution. To mitigate this, we first filter preliminary keypoints via uniform sampling followed by a motion-range-based constraint, which allows us to identify foundational keyframes. These keyframes serve as temporal anchors to partition the full demonstration into a sequence of sub-demonstrations. Within each sub-demonstration, we further refine keypoint selection using the methodology described in ReLAM. During training, the model is supervised to select keypoints and predict subgoals based on the initial state of each sub-demonstration. At inference time, we adopt an iterative execution strategy: upon reaching a predicted subgoal, the anticipation model is re-triggered to perceive the current environment, re-generate keypoints, and predict the next subgoal in a closed-loop fashion. Since each demonstration is associated with a specific textual description, we encode the text using a frozen T5 encoder to obtain a textual representation, which is then provided as an additional input to the anticipation model during training and inference.

Regarding the implementation of IQL for $\pi_0$, we optimize the Q function $Q_\theta(s, a)$ and value function $V_\psi(s)$ using the standard objectives defined in IQL (Kostrikov et al., 2022). Both $Q$ and $V$ share a common architectural backbone: a frozen PaliGemma model (Beyer et al., 2024) is first employed to extract multi-modal representations from visual observations and natural language task instructions. These features are then processed by a Q-Former to produce a fixed-length sequence of latent tokens. The resulting tokens, concatenated with the robot's proprioceptive state vectors and the action embeddings (exclusive to the $Q$-function), are fed into a trainable Transformer-based encoder. We utilize the embedding of the final token as a comprehensive representation, which is then mapped to a scalar value through a Multi-Layer Perceptron (MLP) to produce the estimated $Q$ or $V$ value. After the optimization of the $Q$ and $V$ networks, we compute the advantage $A(s, a) = Q(s, a) - V(s)$ for each transition in the dataset. The policy is then updated using Advantage-Weighted Regression (AWR). Since $\pi_0$ is implemented as a Flow Matching-based policy, we follow the approach of energy-weighted flow matching (Zhang et al., 2025b) to integrate the advantage values into the training objective. Specifically, the advantage $A(s, a)$ is utilized as a per-sample weight to modulate the Flow Matching loss, effectively performing importance sampling to shift the learned distribution towards optimal behaviors. The weighted objective for the policy $\pi_\theta$ can be formulated as:

$$\mathcal{L}(\theta) = \mathbb{E}_{\substack{(s,\mathbf{a}) \sim \mathcal{D} \\ t \sim \mathcal{U}[0,1] \\ \mathbf{a}_t \sim p_t(\mathbf{a}_t | \mathbf{a})}} \left[ \exp\left( \frac{Q(s, \mathbf{a}) - V(s)}{\beta} \right) \| v_\theta(\mathbf{a}_t, s, t) - u_t(\mathbf{a}_t | \mathbf{a}) \|^2 \right], \tag{9}$$

where $\mathbf{a}_t = (1 - t)\mathbf{a}_0 + t\mathbf{a}$ denotes the interpolated action chunk at diffusion time $t$ for a standard optimal transport displacement mapping, and $v_\theta$ is the network trained to regress the multi-step action trajectory, $\beta$ is a temperature parameter that controls the greediness of the policy update.

**B.6. Experiment environments**

We provide a visualization of the experiment environments in Fig. 4.

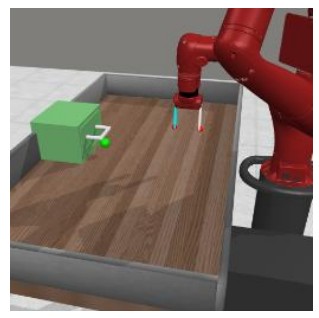 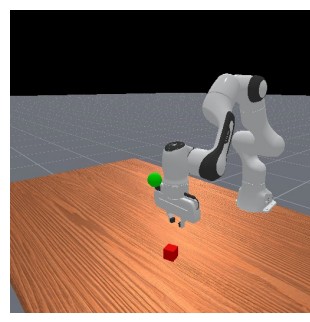 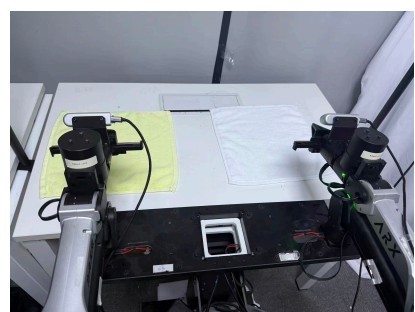

| (A) Meta-World | (B) ManiSkill | (C) ALOHA |

*Figure 4.* A visualization of the environments in our experiments. **(A)** In Meta-World, the agent controls a Sawyer robot to manipulate various objects such as window, drawer and door. **(B)** In the ManiSkill environment, the agent controls a Franka robot with 7-DoF. **(C)** ALOHA robot instructed to fold a towel.

---

**Algorithm 1** Reward Learning with Anticipation Model (ReLAM)

---

**Required**: an action-free video demo dataset $\mathcal{D}$, a text-grounded SAM model EVF-SAM, an off-the-shelf track model Cotracker

**Output**: the optimized robotic control policy $\pi$.

1: Initialize the anticipation model $G_\phi$, policy $\pi_\Phi$, where the subscript denotes their parameters.
2: // Generate keypoint subgoal dataset $\mathcal{K}$
3: **for** each trajectory in $\mathcal{D}$ **do**
4:     Pick out the keypoint in the initial frame with Eq. (1).
5:     Select the key frames using Eq. (2).
6:     Generate the subgoal data with the coordinate of keypoints in key frames.
7: **end for**
8: // Training anticipation model
9: **while** training not converge **do**
10:     Sample keypoint subgoal data from $\mathcal{K}$.
11:     Update $\phi$ by predicting the keypoint sequences with teacher-forcing.
12: **end while**
13: // Training policy
14: **while** policy training not converge **do**
15:     Collect trajectories $(o_t, a_t, o_{t+1}, g_t)$ by rolling out $\pi_\Phi$.
16:     Compute reward for each transition using Eq. (3)
17:     Update $\pi_\Phi$ using collected trajectories with PPO.
18: **end while**
19: **return** the optimized policy $\pi$.

---

## C. Algorithm Description

The practical implementation of ReLAM method for online reinforcement learning is presented in the form of pseudo-code in Algorithm 1.

## D. Mathematical Analysis

We provide a brief mathematical analysis on the effectiveness of ReLAM below. We start by providing an assumption about the learning environment.

**Assumption 1.** *We assume the learning environment, which takes the coordinate of keypoints in the image as state space, satisfies the following conditions:*

*(1) The state space $\mathcal{S}$ is continuous.*

*(2) The state space $\mathcal{S}$ has a Euclidean distance metric $d(\cdot, \cdot)$.*

*(3) The environment's transition function $P$ is deterministic.*

This assumption usually holds in the point space. Apart from this assumption, we impose an additional one that restricts the point's stepwise movement range.

**Assumption 2.** *For each timestep, the robot takes action $a$, and the point $s$ transits to $s'$, which satisfies the following condition:*

*(1) $s'$ is reachable.*

*(2) There exists a fixed constant $M$, s.t. $d(s, s') \leq M$.*

*(3) $\forall \hat{s} \in B(s, M)$, if $\hat{s}$ is reachable, there exists a unique action $\hat{a}$, s.t. $P(s, \hat{a}) = \hat{s}$. For simplicity, we denote $P^{-1}(s, \hat{s}) = \hat{a}$ as the inverse dynamics model.*

Assumption 2 can hold for robotic manipulation environments, especially when the action mode is set to be delta position of the end effector. Next, we will give a definition which describes the motion of robotic manipulation.

**Definition 1** (Linear Reachability). *We say $s'$ is linearly reachable from $s$, if: $\exists \epsilon > 0, \forall \hat{s} \in \mathcal{S}$, if $\exists t \in [0, 1]$, $d(\hat{s}, t \cdot s + (1 - t) \cdot s') < \epsilon$, then $\hat{s}$ is reachable.*

And we define two kinds of reward:

**Definition 2** (Reward definition). *Suppose the final goal is $g$, the we define two reward:*

*(1) Time reward:*

$$r_t(s, g) = \begin{cases} -1, & \text{if } g \text{ is not reached,} \\ 0, & \text{if } g \text{ is reached} \end{cases}$$

*(2) Distance reward:*

$$r_d(s, g) = \begin{cases} f(d(s, g)), & \text{if } g \text{ is not reached,} \\ 0, & \text{if } g \text{ is reached} \end{cases}$$

*where $f$ is a monotonically decreasing function that is always negative.*

With these definitions, we can obtain the following lemma:

**Lemma 1.** *Suppose we have a task starting from $s_0$, and the terminal point state is $g$. The environment satisfies assumption 1, 2. If $g$ is linearly reachable from $s_0$, then the optimal policy $r_d$ is also optimal for $r_t$.*

*Proof.* Based on assumption 2 and linear reachability, it is easy to find that one of the optimal policies for time reward $r_t$ is:

$$\pi_{r_t}^*(a|s) = \begin{cases} P^{-1}(s, s + \frac{M \cdot (g-s)}{d(s,g)}), & \text{if } d(s,g) > M, \\ P^{-1}(s, g), & \text{if } d(s,g) \le M \end{cases} \tag{10}$$

For the distance reward $r_d$, when starting from $s_0$, suppose the episode ends at time $T$, then the total return $G$ is computed as:

$$
\begin{aligned}
G &= \sum_{t=0}^{T} r_d^t(s_t, g) \\
&\le \sum_{t=0}^{T} f(d(s_t, g)) \\
&= \sum_{t=0}^{\lceil d(s,g)/M \rceil} f(d(s_t, g)) + \sum_{t=\lceil d(s,g)/M \rceil + 1}^{T} f(d(s_t, g)) \\
&\le \sum_{t=0}^{\lceil d(s,g)/M \rceil} f(d(s_t, g)) + \sum_{t=\lceil d(s,g)/M \rceil + 1}^{T} 0 \\
&\le \sum_{t=0}^{\lceil d(s,g)/M \rceil} f(d(\hat{s}_t, g))
\end{aligned}
\tag{11}
$$

where $\hat{s}_t = s_0 + t \cdot \frac{M \cdot (g - s_0)}{d(s_0, g)}$ lines in the straight line between $s_0$ and $g$. Eq. (11) shows that the optimal policy for distance reward $r_d$ is the same as Eq. (10). □

Obviously, the policy in Eq. (11) always takes the action to reach the farthest reachable point in the straight line between the current state and the desired goal if it is linearly reachable. Therefore, it defines a shortest path from the start point to the goal point. Next, we will introduce some propositions about the anticipation model and pixel tracking in robotic manipulation.

**Definition 3.** *We say $(g_0, g_1, \cdots, g_k)$ is a path for the start point state $s_0$ and the goal point state $g$, if $g_0 = s_0$, $g_k = g$ and $\forall 0 \le i \le k - 1$, $g_{i+1}$ is linearly reachable from $g_i$. A path is said to be the shortest if $(g_0, \cdots, g_k) = \arg\min \sum_{i=0}^{k-1} d(g_i, g_{i+1})$ among all paths for the task.*

**Assumption 3.** *The learning task has a shortest path $(g_0, g_1, \cdots, g_k)$. And the anticipation model can predict a path $(\hat{g}_0, \cdots, \hat{g}_k)$ satisfying:*

$$\forall 0 \le i \le k, \quad d(\hat{g}_i, g_i) < \epsilon_A \tag{12}$$

With the assumption above, we get $k$ sub-tasks where for each sub-task $i$, the low-level policy $\pi_i^{ReLAM}$ is trained with distance reward using PPO. By denoting the expected return of $\pi_i^{ReLAM}$ under a reward $r$ in sub-task $i$ as $\hat{V}_{i,r}^{\pi_i^{ReLAM}}$, we can obtain the following theorem:

**Theorem 1** (sub-optimality bound for ReLAM). *If for all $0 \le i \le k-1$, the low-level policy $\pi_i^{ReLAM}$ trained with reward $r^{ReLAM}$ satisfies:*

$$|V_{i,r_t}^{\pi_{i,r}^{*}ReLAM} - V_{i,r_t}^{\pi_i^{ReLAM}}| < \epsilon_\pi \tag{13}$$

*which means that the expected time used for $\pi_i^{ReLAM}$ and $\pi_{i,r^{ReLAM}}^*$ to complete the sub-task $i$ is almost the same. Then ReLAM will train a policy $\pi^{ReLAM}$, whose sub-optimality is bounded by:*

$$V_{r_t}^* - V_{r_t}^{\pi^{ReLAM}} \le k \cdot (\epsilon_\pi + \frac{2\epsilon_A}{M}) \tag{14}$$

*Proof.* For all $0 \leq i \leq k-1$, we have:

$$
\begin{aligned}
V_{i,r_t}^{\pi^{ReLAM}} &= V_{i,r_t}^{\pi^{ReLAM}} - V_{i,r_t}^{\pi_{i,r}^{*ReLAM}} + V_{i,r_t}^{\pi_{i,r}^{*ReLAM}} \\
&\geq -\epsilon_\pi + V_{i,r_t}^{\pi_{i,r}^{*ReLAM}} \\
&= -\epsilon_\pi + V_{i,r_t}^{*} \\
&= -\epsilon_\pi + \frac{d(\hat{g}_i, \hat{g}_{i+1})}{M} \\
&\geq -\epsilon_\pi + \frac{d(g_i, g_{i+1}) - 2\epsilon_A}{M}
\end{aligned}
\tag{15}
$$

The first inequality can be deduced by lemma 1 and the fact that $V_{i,r_t}^{\pi_{i,r}^{*ReLAM}} = V_{i,r_t}^{\pi_{r_t}^{*}} = V_{i,r_t}^{*}$. The penultimate line is from the definition of time reward, the max-step-size assumption 2 and the fact that $\hat{g}_{i+1}$ is linearly reachable from $\hat{g}_i$. The last inequality comes from assumption 3 and the triangle inequality. By summing from $i=0$ to $k-1$, we have:

$$
\begin{aligned}
V_{r_t}^{\pi^{ReLAM}} &= \sum_{i=0}^{k-1} V_{i,r_t}^{\pi^{ReLAM}} \\
&\geq \sum_{i=0}^{k-1} \left[ -\epsilon_\pi + \frac{d(g_i, g_{i+1}) - 2\epsilon_A}{M} \right] \\
&= \frac{\sum_{i=0}^{k-1} d(g_i, g_{i+1})}{M} - k \cdot \left( \epsilon_\pi + \frac{2\epsilon_A}{M} \right) \\
&= V_{r_t}^{*} - k \cdot \left( \epsilon_\pi + \frac{2\epsilon_A}{M} \right)
\end{aligned}
\tag{16}
$$

The last equality is because $(g_0, g_1, \cdots, g_k)$ is the shortest path. And finally we can get Eq. (14), which completes the proof. $\qquad \square$

Theorem 1 shows that the learnt policy $\pi^{ReLAM}$ will find an almost shortest path, validating the soundness of our approach.

# E. More Experimental Results

Fig. 5 demonstrates the t-SNE visualization of rewards on the Pick Cube task. Fig. 6 shows the learning curves of some other results on Drawer Open task.

We provide the learning curves for each method on training tasks with online RL in Fig. 7, 8 and curves of offline RL in Fig. 9, 10.

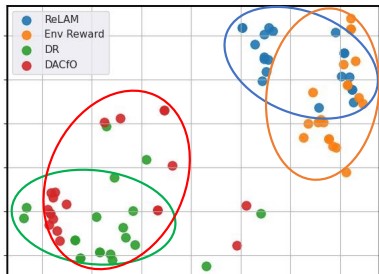

*Figure 5.* The t-SNE visualization of rewards on the Pick Cube task.

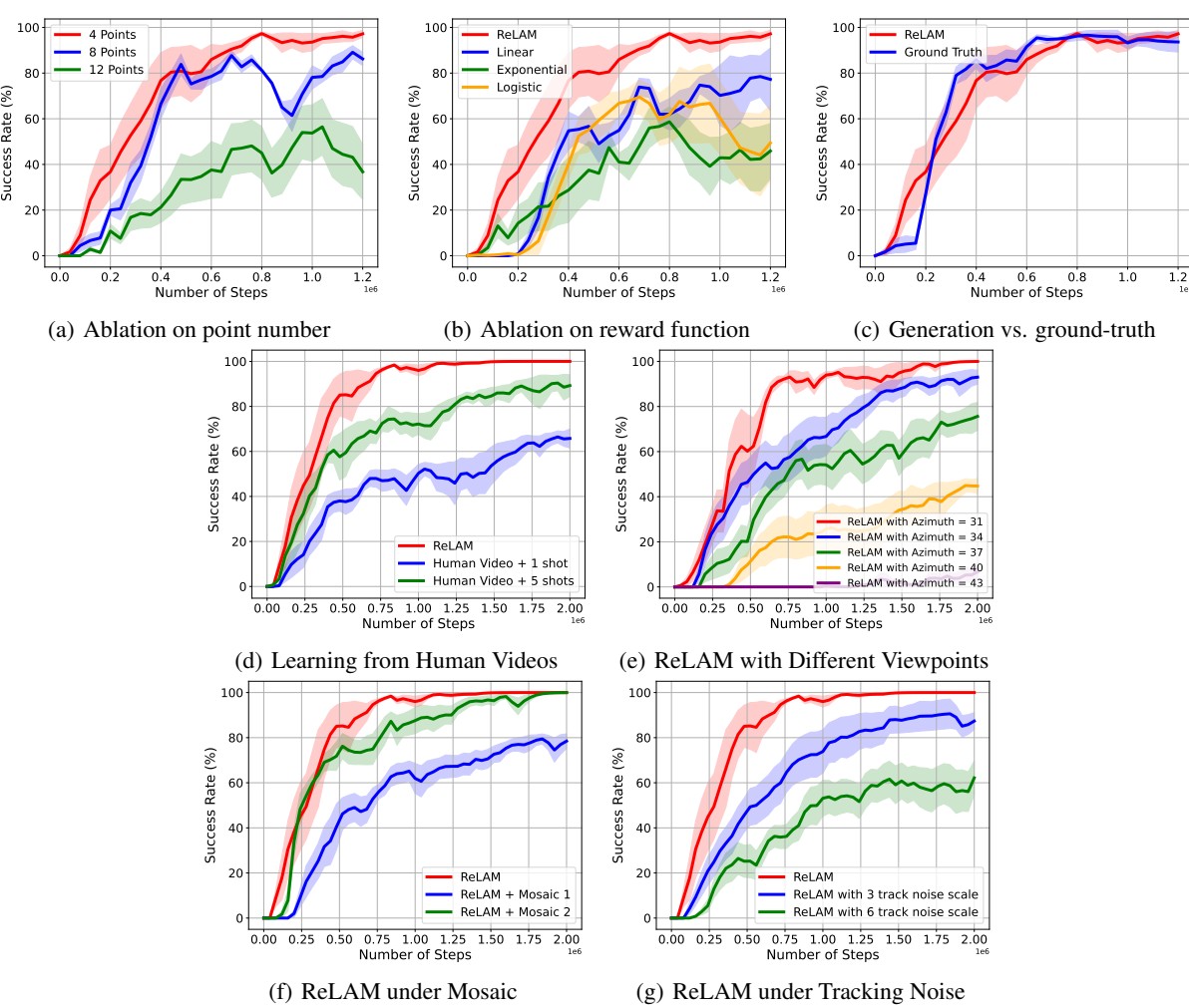

*Figure 6.* Some Other Results on Drawer Open task.

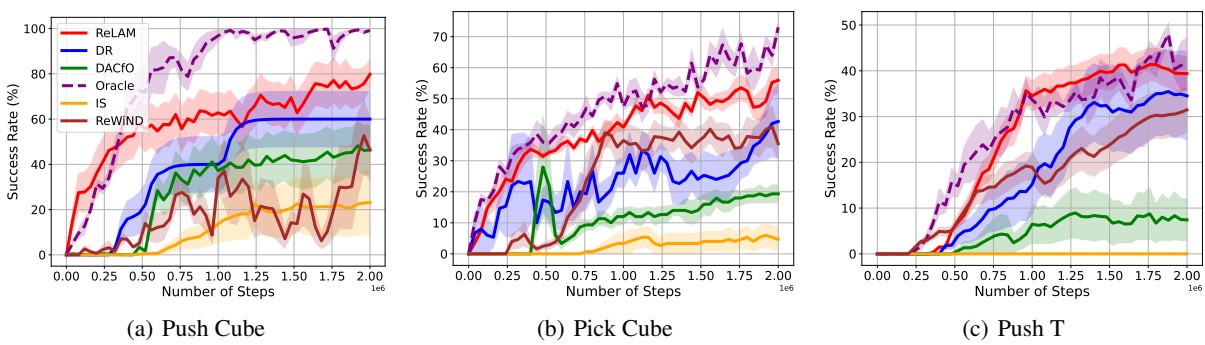

(a) Push Cube

(b) Pick Cube

(c) Push T

*Figure 7.* Learning Curves of Online RL in ManiSkill Tasks

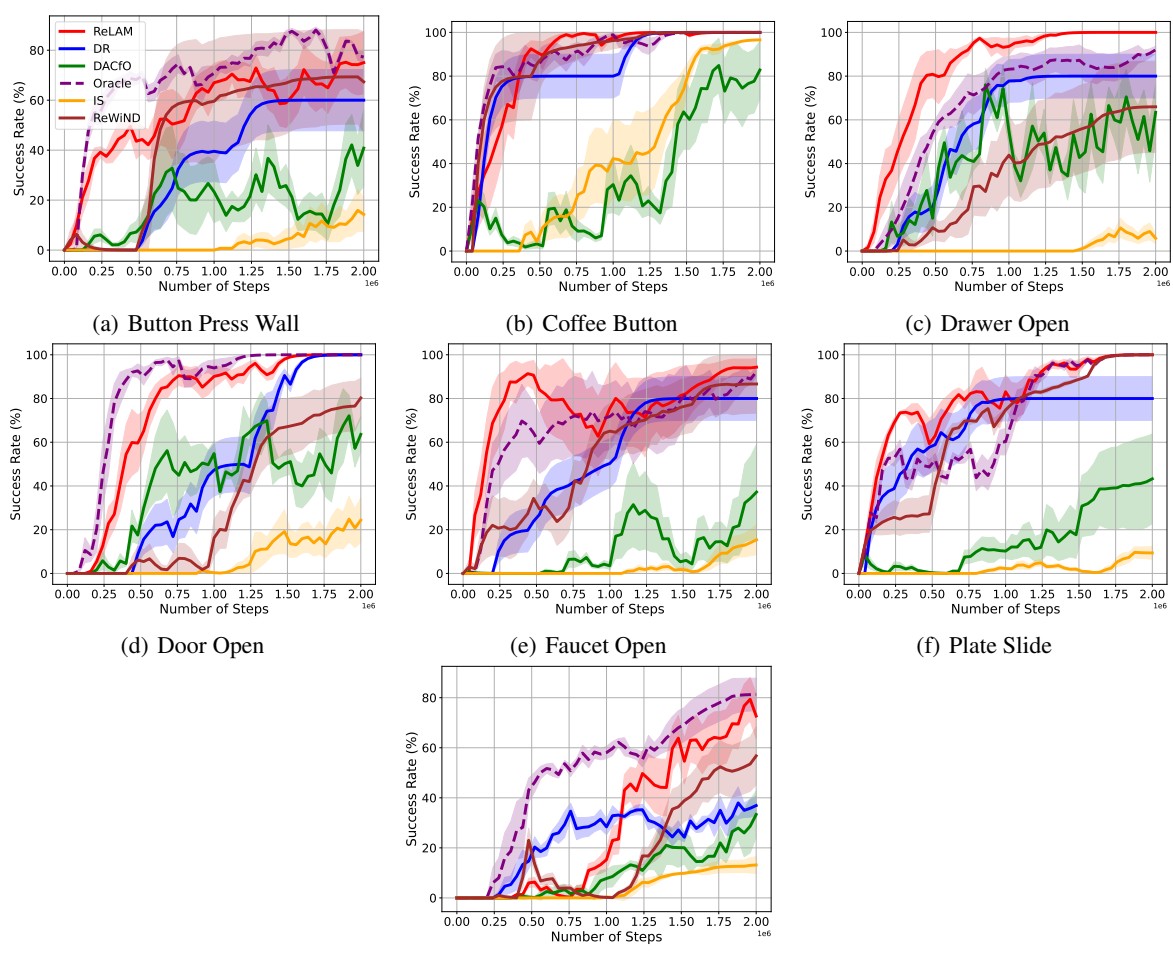

(a) Button Press Wall

(b) Coffee Button

(c) Drawer Open

(d) Door Open

(e) Faucet Open

(f) Plate Slide

(g) Peg Insert Side

*Figure 8.* Learning Curves of Online RL in Meta-World Tasks

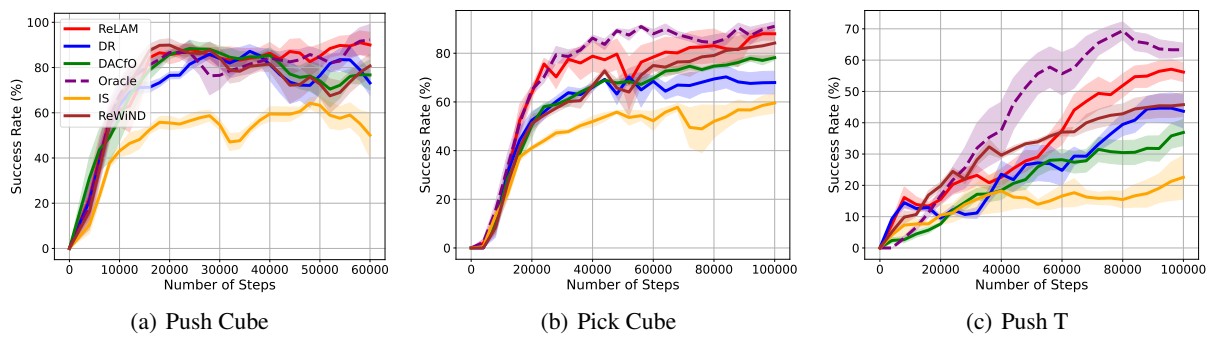

(a) Push Cube  (b) Pick Cube  (c) Push T

*Figure 9.* Learning Curves of Offline RL in ManiSkill Tasks

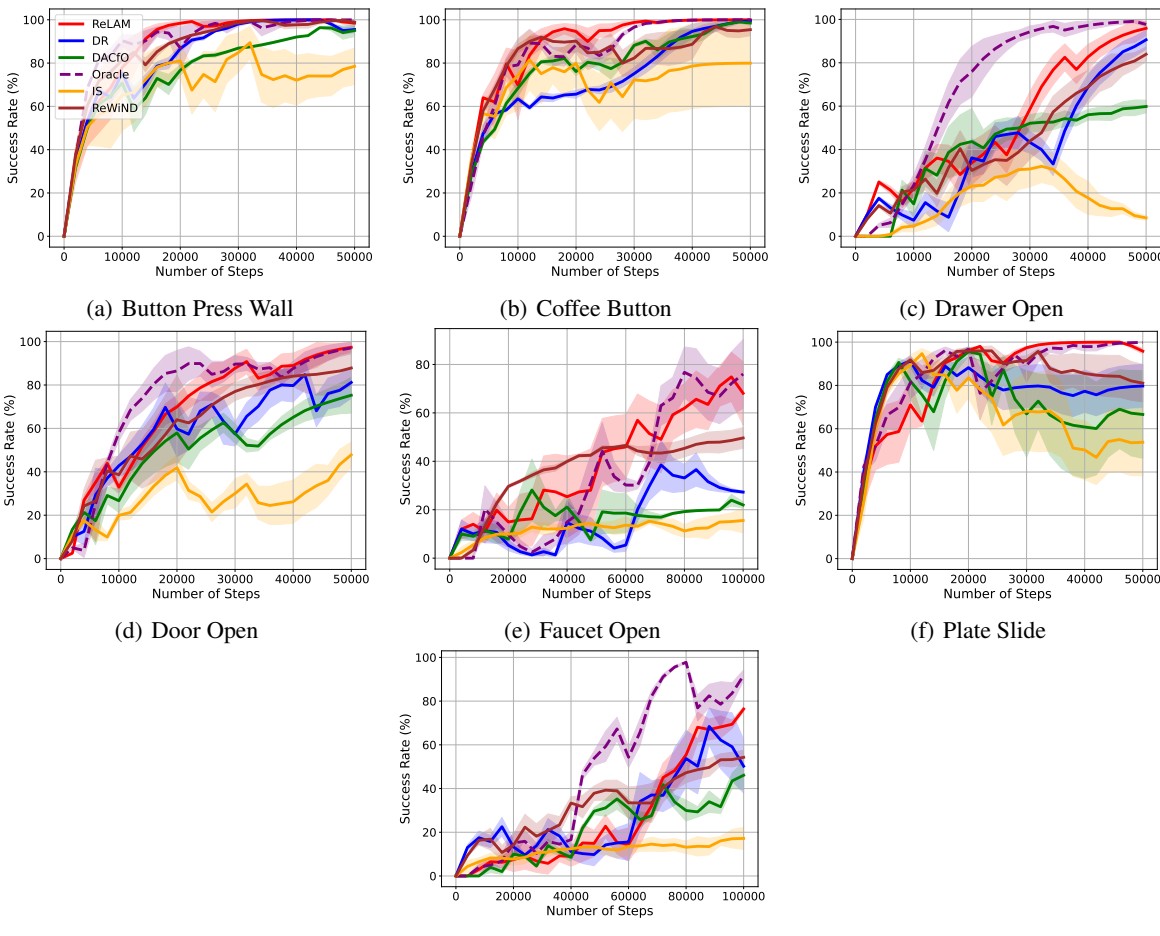

(a) Button Press Wall  (b) Coffee Button  (c) Drawer Open

(d) Door Open  (e) Faucet Open  (f) Plate Slide

(g) Peg Insert Side

*Figure 10.* Learning Curves of Offline RL in Meta-World Tasks

# F. Examples of Generation Results from Anticipation Model

We provide some examples of generated initial keypoints from the anticipation model at both Meta-World, ManiSkill and real-world tasks in Fig. 11. The hollow markers in the figure denote the keypoints generated based on this picture.

Moreover, Fig. 12 demonstrates the results of predicting the next keypoint subgoals using the anticipation model, based upon the current observation images and keypoint locations, across Meta-World, ManiSkill, and real-world tasks. Solid markers denote the current keypoints, while hollow markers indicate the predicted keypoint subgoals according to the solid markers with the same color. Note that in some cases the hollow markers are difficult to distinguish because the predicted points exhibit little displacement relative to their initial positions, resulting in overlap.

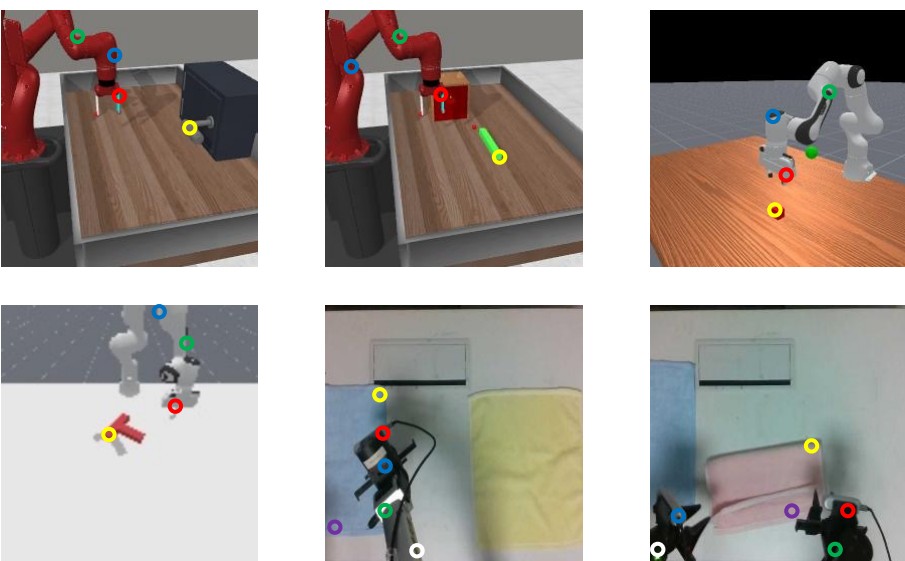

*Figure 11.* Examples of the initial keypoint generated from the anticipation model.

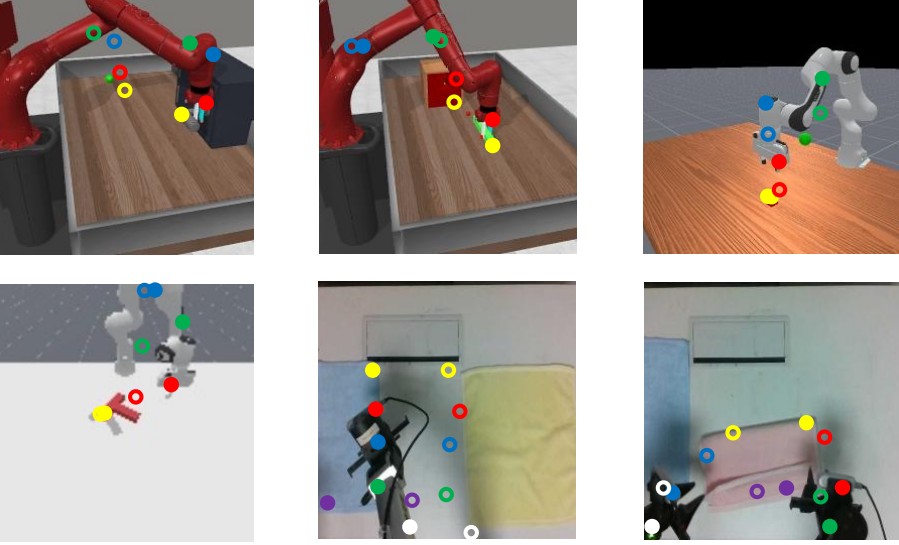

*Figure 12.* Examples of the keypoint subgoal generated from the aniticipation model.

## G. Hyper Parameters

| Hyper-parameters | Value |
|---|---|
| Embedding dimension | 512 |
| Layer Num. | 12 |
| Dropout rate | 0.1 |
| Head Num. | 8 |
| Keypoint Num. | 4 |
| Batch size | 8 |
| Learning rate | $3 \times 10^{-5}$ |

*Table 6.* Core Hyper-parameters for Learning Anticipation Model

| Hyper-parameters | Value |
|---|---|
| Learning rate | $3 \times 10^{-4}$ |
| Batch size | 64 |
| Number of epochs | 10 |
| Gamma | 0.99 |
| GAE lambda | 0.95 |
| Number of Steps | 2000 |
| Clip range | 0.2 |
| Entropy coefficient | 0.0 |
| Value function coefficient | 0.5 |
| Max gradient norm | 0.5 |
| CNN channels | $[16, 32, 64]$ |
| CNN kernel sizes | $[8, 8, 8]$ |
| CNN strides | $[4, 4, 4]$ |
| Mlp hidden dims | $[512, 256]$ |

*Table 7.* Core Hyper-parameters for PPO

| Hyper-parameters | Value |
|---|---|
| Learning rate | $3 \times 10^{-4}$ |
| Batch size | 64 |
| Gamma | 0.99 |
| Tau | 0.005 |
| Expectile | 0.7 |
| Temperature | 3.0 |
| CNN channels | $[16, 32, 64]$ |
| CNN kernel sizes | $[8, 8, 8]$ |
| CNN strides | $[4, 4, 4]$ |
| Mlp hidden dims | $[256, 256]$ |

*Table 8.* Core Hyper-parameters for IQL

## G.1. Example Images of Mosaic

We provide the example images of mosaics on Drawer Open task in Fig. 13.

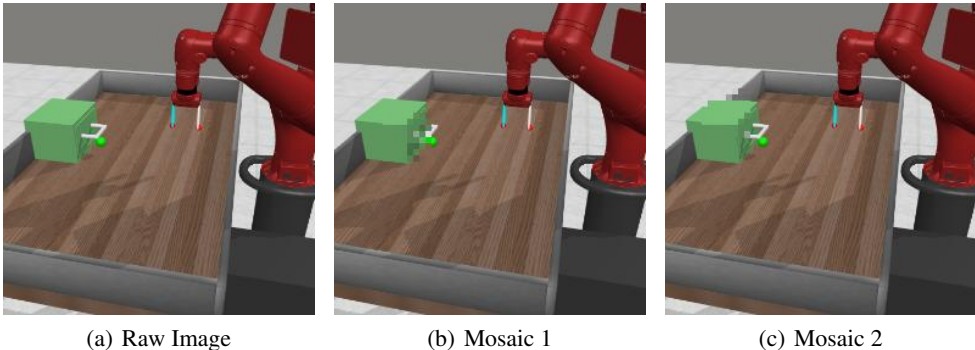

(a) Raw Image          (b) Mosaic 1          (c) Mosaic 2

*Figure 13.* Example images on Drawer Open with: (a) raw image with no mosaic; (b) image with mosaic 1; (c) image with mosaic 2.

