# OpenReview forum: "ReLAM: Learning Anticipation Model for Rewarding Visual Robotic Manipulation"
_ICML.cc/2026/Conference — ICML 2026 regular_

### Official Review · Reviewer_Biku · 2026-02-28

**Soundness:** 3
**Presentation:** 4
**Significance:** 3
**Originality:** 3
**Overall Recommendation:** 5
**Confidence:** 4

**Summary:**

The paper explores the reward design problem for robotic manipulation.

**Compliance With Llm Reviewing Policy:**

Affirmed.

**Key Questions For Authors:**

1. The method requires collecting video demonstrations, which somewhat reduces the advantages of online RL methods that typically do not rely on pre-collected demonstrations.

2. It is impressive that the approach can accomplish long-horizon folding tasks, as such tasks are generally challenging for RL policies to learn. It would be beneficial if the authors could further explain why the proposed method is capable of handling such long-horizon tasks effectively.

**Limitations:**

Refer to above

**Strengths And Weaknesses:**

I really appreciate the core idea of this work:

1. The paper is clearly written, making it easy to follow both the overall workflow and the detailed implementation.

2. The keypoint-based approach for generating dense reinforcement learning rewards is interesting and well-motivated.

3. The experimental evaluation is comprehensive, covering both simulation and real-world settings, as well as both offline and online RL policies.

4. The ablation studies further strengthen the credibility of the proposed method and help validate its feasibility and design choices.

---

> ### Author Rebuttal · Authors · 2026-03-31
>
> We sincerely thank the reviewer for the positive assessment of ReLAM. Below, we respond to each of the comments point by point.
>
> **Q1. Regarding collecting video demonstrations: *The method requires collecting video demonstrations, which somewhat reduces the advantages of online RL methods that typically do not rely on pre-collected demonstrations.***
>
> We believe that ReLAM indeed provides a practical way to leverage expert data in online RL, especially in scenarios where obtaining a dense reward function is difficult. By introducing keypoints, ReLAM incorporates useful spatial priors, and by decomposing the task into a sequence of subgoals, it effectively breaks a complex task into simpler subtasks. This substantially reduces the difficulty of online reinforcement learning.
>
> **Q2. Regarding reasons for handling long-horizon tasks: *It would be beneficial if the authors could further explain why the proposed method is capable of handling such long-horizon tasks effectively.***
>
> We believe ReLAM is able to handle long-horizon tasks for two main reasons:
>
> 1. Automatic task decomposition through a hierarchical perspective: Inspired by the idea of hierarchical reinforcement learning, ReLAM automatically decomposes a long-horizon task into a sequence of subtasks. Compared with requiring the policy to solve the entire task at once, asking it to accomplish these subtasks sequentially is substantially easier.
>
> 2. Compact and structured task representation via keypoints: By extracting task-relevant keypoints, ReLAM obtains a highly condensed abstract representation of the task. These abstractions naturally induce a spatial distance-based reward and can also serve as informative inputs to the policy, thereby facilitating reinforcement learning.
>
> **In addition, we conducted several extra experiments, which we list some below. If you are interested, we would be happy to direct you to these results.**
>
> **For convinience, we have included all experimental results in **[PDF-link](https://anonymous.4open.science/r/kU1mJ2eN1mA3cZ6xT2yZ/rebuttal_materials.pdf)** for reviewer’s reference. In the following, “PDF” refers to this file.**
>
> **Experiment 1: Learning from human videos**
>
> we collected 40 human drawer-opening videos and extracted a keypoint dataset from them using the procedure described in **Section 3.1** of the paper. We found that, starting from this dataset, adding only a very small amount of target-domain keypoint data—namely, keypoint annotations extracted from a small number of expert videos in Meta-World Drawer Open is already sufficient to produce an effective reward. The table below reports the success rates of our method on Meta-World Drawer Open when trained following the ReLAM pipeline with **40 human videos + 1 Meta-World expert video** and **40 human videos + 5 Meta-World expert videos**, with the corresponding training curves are shown in **Fig. 1** of the PDF. These results indicate that ReLAM has strong few-shot transfer capability.
>
> **Experiment 2: Robustness to novel viewpoint**
>
> To study robustness to viewpoint variation, we conducted additional experiments on Meta-World Drawer Open task. Specifically, we sampled 10 camera viewpoints with azimuth angles randomly chosen between **10° and 30°**. For each viewpoint, we collected 10 expert demonstration videos and trained the anticipation model on the resulting 100 videos. We then evaluated policy learning in RL environments with **previously unseen viewpoints**, using azimuth angles of **31°**, **34°**, **37°**, **40°**, and **43°**.
>
> For each of these unseen viewpoints, we collected a small set of ground-truth demonstration videos and extracted a ground-truth keypoint subgoal dataset. We then used the anticipation model to predict keypoint subgoals from the initial frame of each demonstration and measured the average of the L2 distance between the predicted subgoals and the ground-truth subgoals, as reported in the table below:
>
> | Azimuth | 31° | 34° | 37° | 40° | 43° |
> |---------|-----|-----|-----|-----|-----|
> | Prediction error | 2.6 | 7.4 | 12.7 | 28.8 | 67.0 |
>
> We found that, as the test viewpoint gradually deviates from the viewpoints seen during training, the prediction error of the anticipation model also increases. Moreover, once the viewpoint shift exceeds roughly **10°**, the error increases sharply. This suggests that ReLAM exhibits a certain degree of robustness to viewpoint variation.
>
> We also performed RL experiments at the same five unseen viewpoints using the trained anticipation model. As shown in **Fig. 5** of the PDF, the policy success rate drops substantially at an azimuth angle of **43°**, which is consistent with the trend observed in the above table.

---

> > ### Author Rebuttal · Reviewer_Biku · 2026-04-07
> >
> > I still think this is a good paper and tend to accept it

---

> > > ### Author Response · Authors · 2026-04-07
> > >
> > > Thank you for your reply and for recognizing our clarifications and revisions. We will continuously devote to further improving the paper.

---

### Official Review · Reviewer_g9UY · 2026-03-12

**Soundness:** 4
**Presentation:** 3
**Significance:** 3
**Originality:** 3
**Overall Recommendation:** 5
**Confidence:** 2

**Summary:**

This paper addresses a key challenge in vision-based robotic RL that construct informative rewards when precise state information is unavailable. The main idea is to extract task-relevant keypoints from video demonstrations, identify keyframes, train an autoregressive anticipation model to predict intermediate keypoint subgoals, and then use distance to those predicted subgoals as dense rewards for low-level policy learning. The experimental evaluation is broad, covering both online and offline RL on Meta-World and ManiSkill manipulation tasks, as well as a real-world ALOHA towel-folding setup. Overall, the results are encouraging, particularly given the diversity of the evaluation and the comparisons against several strong video-based reward-learning baselines.

**Compliance With Llm Reviewing Policy:**

Affirmed.

**Final Justification:**

During the rebuttal, the authors provided detailed responses and additional experiments that adequately addressed my concerns.

**Key Questions For Authors:**

- Can you directly evaluate the anticipation model, beyond downstream success, using prediction error or robustness under occlusion/view changes?
- How much task-specific tuning is required for prompts, thresholds, and camera selection? Do shared defaults work across tasks?
- How sensitive is performance to segmentation/tracking errors, such as drifting or missing keypoints?
- On the two hardest online tasks, how much of the gain comes from ReLAM itself versus the added expert demos and BC loss?

**Limitations:**

yes

**Strengths And Weaknesses:**

## Strengths
- The method design is reasonable. Keypoint extraction, keyframe selection, autoregressive subgoal generation, and keypoint-distance reward shaping fit together naturally within the HRL formulation.
- The evaluation is broad for this type of work, spanning online RL, offline RL, multiple simulation benchmarks, and a real-world ALOHA towel-folding experiment rather than a single narrow setting.
- The ablations are meaningful and support important design choices, including the number of keypoints, the reward function shape, and generated versus ground-truth subgoals.
- The real-world ALOHA result is small-scale, but it strengthens the claim that the approach may transfer beyond simulation and be useful in practical robot-learning pipelines
- The paper clearly frames reward design as the central challenge, then introduces ReLAM as a two-stage system

## Weaknesses
- Two hard online tasks use 10 expert demos with a BC auxiliary loss, which may weak the paper's "action-free video reward learning" claim on the most difficult settings.
- The method seems most suitable for tasks whose progress is well captured by image-space keypoints; it is less convincing for heavy occlusion, deformation, or contact-rich manipulation.
- Despite the "automatic" framing, the method still appears to require nontrivial task-specific setup (for object prompts, thresholds, and camera/view choices), and the paper does not clearly quantify this burden.

---

> ### Author Rebuttal · Authors · 2026-03-31
>
> Thank you for your thorough review and for providing many valuable comments that have helped improve the quality of our manuscript. In the following, we address each of your comments point by point.
>
> **For convinience, we have included all experimental results in **[PDF-link](https://anonymous.4open.science/r/kU1mJ2eN1mA3cZ6xT2yZ/rebuttal_materials.pdf)** for reviewer’s reference. In the following, “PDF” refers to this file.**
>
> **Q1. Regarding robustness under view change: *Can you directly evaluate the anticipation model under view changes?***
>
> To study robustness to viewpoint variation, we sampled 10 camera viewpoints with azimuth angles randomly chosen between **10° and 30°** on Drawer Open task. For each viewpoint, we collected 10 expert demonstration videos and trained the anticipation model on the resulting 100 videos. We then evaluated policy learning in RL environments with **previously unseen viewpoints**, using azimuth angles of **31°**, **34°**, **37°**, **40°**, and **43°**.
>
> For each of these unseen viewpoints, we collected a small set of ground-truth demonstration videos and extracted a ground-truth keypoint subgoal dataset. We then used the anticipation model to predict keypoint subgoals from the initial frame of each demonstration and measured the average of the L2 distance between the predicted subgoals and the ground-truth subgoals, as reported in the table below:
>
> | Azimuth | 31° | 34° | 37° | 40° | 43° |
> |---------|-----|-----|-----|-----|-----|
> | Prediction error | 2.6 | 7.4 | 12.7 | 28.8 | 67.0 |
>
> We found that, as the test viewpoint gradually deviates from the viewpoints seen during training, the prediction error of the anticipation model also increases. Moreover, once the viewpoint shift exceeds roughly **10°**, the error increases sharply. This suggests that ReLAM exhibits a certain degree of robustness to viewpoint variation.
>
> We also performed RL experiments at the same five unseen viewpoints using the trained anticipation model. As shown in **Fig. 5** of the PDF, the policy success rate drops substantially at an azimuth angle of **43°**, which is consistent with the trend observed in the above table.
>
> **Q2. Regarding task-specific tuning: *How much task-specific tuning is required?***
>
> 1. Prompt design for task-relevant segmentation: Thanks to recent advances in grounding-based segmentation models, we do not need to spend substantial effort on engineering prompts for segmentation.
> In practice, we only need to specify the task-relevant object to obtain the corresponding segmentation. For example, in the Door Open task, using the prompt **“door”** is sufficient, and for segmenting the manipulator, the prompt **“robot arm”** works well. For objects that are harder to describe precisely, such as the small cube in Push Cube, adding the object’s color is enough: the prompt **“blue cube”** enables SAM to obtain the correct segmentation. We do not view this as a practical obstacle. Moreover, as grounding-based SAM models continue to improve, obtaining accurate segmentations should only become easier and more reliable.
>
> 2. Threshold for subgoal completion: For determining whether a subgoal has been achieved, we use a fixed threshold of **6** for all tasks. We also provide a thorough form of reward we use in the **Q1** of rebuttal for reviewer qNRY.
>
> 3. Camera selection: For Meta-World tasks, we use observation viewpoints with azimuth angles of **12.5°** or **167.5°**, since certain tasks suffer from severe occlusion under some camera views; you can see **Appendix B.1** in the paper for details. For ManiSkill tasks, we use the default camera viewpoints provided by the official environment.
>
> **Q3. Regarding segmentation and tracking errors:*How sensitive is performance to segmentation/tracking errors?***
>
> We design two experimental settings to evaluate the sensitivity of ReLAM to segmentation and tracking errors. Please refer to **Q2** of the rebuttal to reviewer ZoxA.
>
> **Q4. Regarding extra gain from BC: *how much of the gain comes from ReLAM itself versus the added expert demos and BC loss?***
>
> We conducted pure BC experiments on both Peg Insert Side and Pick Cube task using 100 expert trajectories with actions. For fairness, the BC policy also used a CNN backbone and was trained for 200,000 update steps with a batch size of 128. This setup ensures that BC and ReLAM trained with PPO observe a comparable number of state-action pairs, both at about $2 \\times 10^7$. The results, shown in **Fig. 6** of the PDF, indicate that even when action chunking (chunk size = 8) is used to mitigate BC’s compounding error, pure BC still performs worse than **ReLAM + a small amount of expert BC data** (10 expert trajectories). We think the BC loss during training of ReLAM policy can serve as a good initialization for RL and as a useful regularizer during training.

---

> > ### Author Rebuttal · Reviewer_g9UY · 2026-04-05
> >
> > Thank you for the detailed responses and additional experiments, which have adequately addressed my concerns. With these clarifications and improvements, I believe the paper is now ready. I have increased my score.

---

> > > ### Author Response · Authors · 2026-04-06
> > >
> > > We sincerely thank the reviewer for taking time to read our clarifications and for reconsidering the assessment. We are glad that the rebuttal addressed your concerns. Thank you again for your constructive engagement throughout the review process.

---

### Official Review · Reviewer_ZoxA · 2026-03-17

**Soundness:** 2
**Presentation:** 2
**Significance:** 2
**Originality:** 2
**Overall Recommendation:** 3
**Confidence:** 5

**Summary:**

This paper studies reward design for vision-based robotic manipulation and proposes ReLAM, a framework that learns dense rewards from action-free video demonstrations. The method extracts and tracks task-relevant image keypoints, selects keyframes to form intermediate subgoals, and trains an anticipation model to predict these subgoals autoregressively; a low-level goal-conditioned policy is then optimized using rewards based on the distance between current and predicted keypoints within a hierarchical reinforcement learning framework. The main contributions are a keypoint-based reward formulation, the use of an anticipation model to generate structured subgoal from videos without action labels, and empirical validation on Meta-World, ManiSkill, and a real-world ALOHA towel-folding task showing improved performance on long-horizon manipulation tasks.

**Compliance With Llm Reviewing Policy:**

Affirmed.

**Final Justification:**

I increased my overall evaluation to 3

**Key Questions For Authors:**

1) Can you compare ReLAM against a 3D baseline when RGB-D or multi-view sensing is available? For rigid-object tasks, a baseline using 3D pose trajectories or 3D subgoals would help clarify when 2D keypoint-based rewards are actually preferable. A strong result here would improve my view of the paper’s practical significance.

2) How robust is the method to featureless or visually ambiguous objects? Since ReLAM depends on segmentation and keypoint tracking, it would be helpful to discuss failure cases and sensitivity to tracking noise on low-texture objects.

3) How does the method handle camera motion or viewpoint change during manipulation? This is important for settings such as humanoids, mobile manipulators, or wrist-mounted cameras, where the observation frame is not fixed. Evidence of robustness, or a clear discussion of how to extend the method, would strengthen my assessment of generalizability.

**Limitations:**

The authors do acknowledge several technical limitations, including viewpoint dependence, occlusion sensitivity, and limited dataset/model scale, which is appreciated. It would strengthen the paper to explicitly discuss sensitivity to reliable segmentation/keypoint tracking, likely difficulty on featureless or visually ambiguous objects, and the reduced utility of the approach in settings where accurate 3D state is already available from RGB-D or multi-view perception.

**Strengths And Weaknesses:**

Soundness: The paper is technically coherent and the overall method is well motivated. the hierarchical design, keypoint-based subgoals, and evaluation across simulation and a real ALOHA towel-folding task provide reasonable empirical support for the main claims. At the same time, the evidence is somewhat limited by strong assumptions about reliable segmentation, keypoint tracking, and viewpoint consistency, and the theoretical analysis appears to rely on simplified conditions that do not fully capture practical manipulation settings with occlusion, ambiguous appearance, or unavailability of sensor-rich 3D perception which is nowadays quite commonly available.

Presentation: The paper is generally clear and easy to follow, with a clean decomposition into keypoint extraction, anticipation modeling, and low-level policy learning, and the experimental section gives a readable picture of how the method is used in practice. However, the paper could do a better job clarifying its operating regime and limitations upfront, especially its dependence on fixed viewpoints, stable visual tracking, and task-specific demonstration videos, as these constraints are important for understanding when the method is likely to succeed.

Significance: Unfortunately I am not seeing much significance in this work. The premise on which the paper is written is to some extent made up. For example unavailability of 3D point cloud or multi-view information which are now common place in robotic settings. As such, the practical significance is somewhat narrowed  and the proposed 2D keypoint formulation may be less compelling.

Originality: The work presents a meaningful and well-integrated combination of ideas, but leads to partially solving a problem that maybe should not be solved at first place. Not to mention many challenges that are still not addressed such as reliable tracking of keypoints accross frames, the issue of occlusion where certain keypoints may appear and disapear etc.

---

> ### Author Rebuttal · Authors · 2026-03-31
>
> Thank you for your thorough review. In the following, we address each of your comments point by point.
>
> **For convinience, we have included all experimental results in **[PDF-link](https://anonymous.4open.science/r/kU1mJ2eN1mA3cZ6xT2yZ/rebuttal_materials.pdf)** for reviewer’s reference. In the following, “PDF” refers to this file.**
>
> **Q1. Regarding 3D baseline：*A baseline using 3D pose trajectories or 3D subgoals would help.***
>
> We adapted the ReLAM setup on top of SPOT to construct a 3D baseline. After obtaining the task-relevant object pose trajectory, we followed the same high-level procedure as ReLAM: we selected a set of keyframes and used their poses as intermediate subgoals. We then trained a diffusion model to predict future pose subgoals, which were used during RL both to provide subgoal supervision and to compute rewards using distance between the current pose and target pose. Since object meshes are more readily available in ManiSkill, we conducted experiments on the ManiSkill Push Cube task. As shown in **Fig. 2** of the PDF, this adapted 3D version indeed improves over the original ReLAM baseline. We think that the improvement comes from the fact that 3D pose information helps the policy localize the object more accurately; with only a single 2D view, the gripper can sometimes mistakenly infer that it is aligned with the object when it is not.
>
> However, as the reviewer pointed out in Key Question 3: “How does the method handle camera motion for settings such as wrist-mounted cameras?”, we also made a simple modification to the original single-view ReLAM by incorporating a wrist-mounted view. For the third-person view, we continued to train the anticipation model and assign rewards in original manner. The wrist view was primarily used to determine whether the gripper was approaching the object and whether it was properly aligned with it. Because the cube is blue, we used a simple heuristic based on the presence of a sufficient number of blue pixels to determine whether the object was visible in the image. We adopted this direct implementation for simplicity because object visibility detection is a relatively mature capability that many modern VLM-based systems can support. When the object was not visible, we assigned a large penalty reward. When the object was visible, we identified the center pixel of the object, measured the distance between that pixel and the gripper pixel, and assigned rewards based on this distance. Policies trained with this reward design outperformed (seen in **Fig. 2** of the PDF) both the original single-view ReLAM and SPOT. We believe this is because, once the wrist view resolves the alignment issue, predicting 2D keypoints becomes easier and more reliable than predicting full 3D poses, both in terms of forecasting difficulty and tracking robustness.
>
> References:
>
> [1] Hsu et al. SPOT: SE(3) Pose Trajectory Diffusion for Object-Centric Manipulation. ICRA 2025
>
> **Q2. Regarding sensitivity to tracking error:  *Discuss sensitivity to tracking noise on low-texture objects***
>
> We design two settings to show the sensitivity to track error.
>
> **Setting 1.** On the Meta-World Drawer Open task, we introduced mosaic perturbations on the drawer as a form of visual distraction. We considered two types of mosaics(**Fig. 7** of PDF). In the 1st setting, the mosaic occludes most of the handle region. In the 2nd setting, the mosaic coveres the rear portion of the drawer, not occluding the handle. In both cases, the mosaic size was fixed at 40 × 40 pixels. For expert demonstration videos, we applied each type of mosaic separately and trained the anticipation model following ReLAM procedure. During RL, we applied the corresponding mosaic to the environment images used for policy training. The results(**Fig. 3** of the PDF) indicate that the 2nd mosaic has almost no effect on policy performance, since the occluded region does not include the drawer handle. In contrast, under the 1st mosaic, the handle is fully occluded in some frames, which can cause CoTracker to fail. When the handle is not completely occluded, ReLAM remains effective.
>
> **Setting 2.** On the Drawer Open task, we further studied the effect of tracking error by adding Gaussian noise of different magnitudes to the coordinates of the points tracked by CoTracker in the RL environment. The results (**Fig. 4** of the PDF) shows that larger scale of tracking noise leads to a greater degradation in policy performance. At the same time, we always uses the keypoints from the initial frame as the reference for Cotracker, which prevents error accumulation over time. This property helps explain why ReLAM remains relatively robust to moderate tracking noise.
>
> **Q3. Regarding Viewpoint Change: *How does the method handle viewpoint change?***
>
> Please refer to **Q1** of the rebuttal to reviewer g9UY.
>
> **Q4. Regarding Wrist-mounted Camera: *This is important for settings such as wrist-mounted cameras.***
>
> See **Q1**.

---

> > ### Author Rebuttal · Reviewer_ZoxA · 2026-04-03
> >
> > The rebuttal is helpful and strengthens the empirical case, especially through additional 3D and tracking-noise experiments. However, it does not fully resolve my central concern that the method’s practical value is limited to a relatively narrow regime where reliable 3D state is unavailable and camera viewpoints remain sufficiently stable. The added wrist-camera heuristic is promising, but it does not yet demonstrate general robustness to viewpoint change or camera motion. Thus, while my concerns about soundness are somewhat reduced, my concerns about significance and generalizability remain.I increased my overall recommendation.

---

> > > ### Author Response · Authors · 2026-04-06
> > >
> > > Thank you for the reply and raising the score. We answer your follow-up question as follow.
> > >
> > > **Q1. The significance and generalizability of ReLAM.**
> > >
> > > To clarify our intended claim, the significance of ReLAM is not that it solves the most general visual manipulation setting with arbitrary camera motion. Rather, we focus on an important and practically common regime in which precise 3D state is unavailable, while camera viewpoints are reasonably stable. In this regime, **reward design remains a major bottleneck, and our contribution is to show that keypoint-based anticipated subgoals can provide dense, structured rewards** from action-free videos (as support by the comment from Reviewer Biku). We believe the empirical evidence supports this claim, i.e., ReLAM consistently improves over prior video-based reward learning baselines across multiple Meta-World and ManiSkill tasks, and also transfers to a real-world ALOHA towel-folding setup. **The additional 3D baseline further suggests that the overall anticipation-and-subgoal recipe is not tied to a single 2D implementation, while the wrist-camera result suggests a possible path toward handling more challenging viewpoints**.
> > >
> > > At the same time, we agree that broader viewpoint robustness remains an open limitation of the current paper. This is also consistent with the limitation discussed in the manuscript: our current anticipation/tracking pipeline depends on relatively stable viewpoints, and a more principled extension would likely require multi-view fusion or 3D point-cloud representations.
> > >
> > > ---
> > >
> > > We appreciate this point, and we hope the paper can be evaluated as a contribution toward scalable reward learning in the "visual observations without reliable 3D state" regime, while leaving unstable viewpoint extensions to future work.

---

### Official Review · Reviewer_qNRY · 2026-03-17

**Soundness:** 2
**Presentation:** 2
**Significance:** 2
**Originality:** 2
**Overall Recommendation:** 3
**Confidence:** 4

**Summary:**

This paper is focused on reward engineering in the domain of vision-based robot manipulation hierarchical RL when action-free video demonstrations are provided. The authors propose Reward Learning with Anticipation Model (ReLAM), taking inspiration from the prior work ATM which shows that predicting future keypoint positions can guide robot policy. Unlike ATM, which performs behavior cloning, ReLAM predicts the keypoints for keyframes, and between keyframes use pixel distance to create a dense reward signal that guides RL learning along the keyframes.

The ReLAM framework consists of the following components:
1. Keypoint selection: grounded SAM gives the open vocabulary segmentation masks of the objects of interest and a tracking model tracks the pixels across the video. From the all segmented pixels, only those which incur large motion, indicating task relevancy, are kept. Finally the authors use furthest point sampling to pick the keypoints.
2. Keyframe selection: within a window of frames, the one at which the keypoint movements turn the biggest angle on average is chosen as the key frame.
3. Anticipation Model (conditional keypoint trajectory generation): similar to ATM, but the authors only predict the keyframe keypoints. A causal transformer conditions the prediction of keypoint trajectories on the initial image observation, encoded with a frozen DINOv2 model.
4. RL with dense reward: the authors assume the visual changes to be simple between keyframes, and hence a keypoint pixel L2 distance is sufficient. Going from one stage to the next is determined when the average distance from the current keypoints to the subgoal is below a threshold. Then they use a piecewise linear reward function to train the RL policy.

Empirical evaluations are done in simulated Meta-World and ManiSkill environments, and on a towel folding task with a real ALOHA robot. ReLAM shows superior task evaluation performance in both online and offline RL. Ablations justify the reward function design, and a t-SNE visualization show the ReLAM rewards projected closer to the environment reward based on object distances.

**Compliance With Llm Reviewing Policy:**

Affirmed.

**Final Justification:**

Raised my score from 2 to 3.

The rebuttal provided missing details on reward function design. The BC results are convincing. Human video + few shot experiment shows initial promising results in a more realistic problem setting.

**Key Questions For Authors:**

Please see above in weaknesses.

- For evaluation of the different stages of the cloth folding task on ALOHA, do you perform the evaluation in one go, regardless of if the previous stage has failed? From the gif it appears to be the case but please confirm.

**Limitations:**

Yes.

**Strengths And Weaknesses:**

Strengths:
1. The core ReLAM framework is explained clearly.
2. Empirical results support ReLAM as a superior reward shaping method compared to existing work.

Weaknesses:
1. Some of the discussions around the reward design are sloppy. For example $r_\mathrm{success}$ appears in equation 3 but is never defined. Equation 4 is supposed to be one of the key technical details of this paper as an reward engineering approach, yet it appears in the appendix and fails to define the variables. Also what are the exact equations of the other rewards in comparison? Also what is $I_\mathrm{task}$?
2. This approach require quite a few (100) demonstrations, at which point even simple BC could have a pretty good performance. Yes the video demonstrations don't have robot actions, and I find this problem setting artificial. Under what circumstances would someone have video-only demonstrations with robot embodiments? With the common EEF pose action space, it is easy to derive the action from states in the simulation. One case where video-only demonstrations do make sense is if it is a human demonstrator. But this paper doesn't show any of those results.

---

> ### Author Rebuttal · Authors · 2026-03-31
>
> Thank you for your thorough review and for providing many valuable comments that have helped improve the quality of our manuscript. In the following, we address each of your comments point by point.
>
> **For convinience, we have included all experimental results in **[PDF-link](https://anonymous.4open.science/r/kU1mJ2eN1mA3cZ6xT2yZ/rebuttal_materials.pdf)** for reviewer’s reference. In the following, “PDF” refers to this file.**
>
> **Q1. Regarding reward design: *Some of the discussions around the reward design are sloppy.***
>
> We thank the reviewer for pointing this out. We agree that the original discussion of the reward design was not sufficiently precise, and we provide the full specification below.
>
> For the original ReLAM reward, we use the following piecewise function:
>
> $$
> r(d)=
> \\begin{cases}
> 0, & d \leq 6,\\\\
> -0.05d + 0.3, & 6 < d \\leq 14,\\\\
> -0.025d - 0.05, & 14 < d \\leq 30,\\\\
> -0.01d - 0.5, & d > 30.
> \\end{cases}
> $$
>
> Here, $d$ denotes the average distance between the current keypoints and the target keypoints. We use this reward formulation for all tasks in both Meta-World and ManiSkill. We set the reward to 0 when $d \leq 6$ because in the actual experiment we set the success threshold as 6, meaning the subgoal is reached.
> For the ablation study on the number of keypoints, we also use the same formula above.
>
> For the ablation study comparing different reward formulations, we use the following alternatives. For the *linear reward*, we use:
>
> $$
> r(d)=
> \\begin{cases}
> 0, & d \\leq 6,\\\\
> -\\frac{d-6}{30}, & d > 6.
> \\end{cases}
> $$
>
> For the *logarithmic reward*, we use:
>
> $$
> r(d)=
> \\begin{cases}
> 0, & d \\leq 6,\\\\
> -0.8 \\cdot \\frac{\\log d - \\log 6}{\\log 5}, & d > 6.
> \\end{cases}
> $$
>
> For the *exponential reward*, we use:
>
> $$
> r(d)=
> \\begin{cases}
> 0, & d \\leq 6,\\\\
> -0.8 \\cdot \\frac{1 - e^{6-d}}{1 - e^{-24}}, & d > 6.
> \\end{cases}
> $$
>
> We would also like to clarify the notation $I_{\text{task}}$ in the paper. We sincerely apologize for the inaccurate description in the submission. In our actual implementation, the anticipation model takes the *initial frame of the task* as input to predict the keypoint subgoals. We thank the reviewer for catching this error.
>
> In addition, during RL training, the reward used in practice consists of three components. The first component, $r_{\text{dense}}$, is the dense reward computed from the keypoint distance as defined above. The second component is a sparse success reward, which is positive only when the entire task is successfully completed; in our experiments, this value is set to $10$, and it is $0$ otherwise. The third component is a subgoal-completion reward, which determines whether the current subgoal has been reached. Specifically, when the distance $d$ is below a predefined threshold $\theta$, we assign an additional positive reward; as said above, we use $\theta = 6$ and set this reward to $5$. Otherwise, this term is $0$. Therefore, the final reward is:
>
> $$
> r_{\text{final}}(d)=r_{\text{dense}} + 10 \cdot \mathbf{1}(\text{success}) + 5 \cdot \mathbf{1}(d \leq 6).
> $$
>
>
> **Q2. Regarding problem setting: *One case where video-only demonstrations do make sense is if it is a human demonstrator.***
>
> We thank the reviewer for this insightful comment. To investigate this setting, we collected 40 human drawer-opening videos and extracted a keypoint dataset from them using the procedure described in **Section 3.1** of the paper. We found that, starting from this dataset, adding only a very small amount of target-domain keypoint data—namely, keypoint annotations extracted from a small number of expert videos in Meta-World Drawer Open is already sufficient to produce an effective reward. The table below reports the success rates of our method on Meta-World Drawer Open when trained following the ReLAM pipeline with **40 human videos + 1 Meta-World expert video** and **40 human videos + 5 Meta-World expert videos**, with the corresponding training curves are shown in **Fig. 1** of the PDF. These results indicate that ReLAM has strong few-shot transfer capability.
>
> The reviewer also noted that “at which point even simple BC could have a pretty good performance.” To examine this directly, we trained a visual **BC baseline** on 100 expert trajectories with actions for Meta-World Drawer Open task, Meta-World Peg Insert Side task and ManiSkill Pick Cube Task. For a fair comparison, we used the same CNN backbone for the BC policy and trained it with 128 batch size for 200000 steps, making it trained with almost the same number of $s,a$ pair (including repeated sample) as ReLAM. The results (as displayed in **Fig. 6** of the PDF) show that BC achieves worse performance than ReLAM even when action chunking (chunk size = 8) is applied to reduce the compounding error of BC. This suggests that the benefit of our method is not merely due to access to a small amount of target-domain data, but also comes from the effectiveness of the reward induced by ReLAM.

---

> > ### Author Rebuttal · Reviewer_qNRY · 2026-04-04
> >
> > Thank you to the authors for the detailed rebuttal and the new experiments. I appreciate the effort you put into this, and I will be raising my score to a 3 in light of these updates.
> >
> > Specifically, the following points are addressed:
> >
> > - The corrected reward formulations.
> > - Gathering human videos and showing successful few-shot transfer to the robot domain was an good addition. This is a good start to addresses my concern about the artificial nature of the problem setting.
> > - The BC baseline demonstrates that ReLAM is better.
> >
> > That being said, fully integrating all of these critical updates will require a fairly substantial revision to the paper's writing.
> >
> > Finally, it looks like one of my questions was missed: For the evaluation of the different stages of the cloth folding task on ALOHA, do you perform the evaluation in one go, regardless of whether the previous stage has failed? I would appreciate a quick clarification on this."

---

> > > ### Author Response · Authors · 2026-04-04
> > >
> > > We sincerely thank the reviewer for the valuable suggestions. Below, we respond to the questions in turn.
> > >
> > > **Q1. Regarding paper revision.**
> > >
> > > We respectfully believe that the suggested improvements **do not change the core philosophy of ReLAM**: ReLAM learns an anticipation model  from action-free video demonstrations that  provides the policy with robust, generalizable, and transferable subgoal and reward signal. For example, the additional experiment we provided to reviewer qNRY on learning the anticipation model from human videos demonstrates the few-shot transfer capability of ReLAM. Likewise, the additional experiments we provided to reviewer ZoxA, including the multi-view, mosaic perturbation, and tracking error studies, demonstrate the robustness of ReLAM to visual disturbances. In our view, these additions further strengthen the central claims of the paper and highlight the broader potential of ReLAM for embodied AI.
> > >
> > > At the same time, we agree that incorporating the reviewer’s suggestions into the paper will **only require some reorganization of the manuscript**. Taking reviewer qNRY’s comments as an example, we plan to revise the paper structure as follows:
> > >
> > > (1) We will provide the full and correct reward definition in **Appendix B.2**.
> > >
> > > (2) We will add the experiment on learning from human videos to **Section 4.2**, between the current *Real-world Evaluation* and *Some Other Results* parts.
> > >
> > > (3) We will include the comparison to the BC baseline in **Section 4.3** as an ablation study.
> > >
> > > **Q2. Regarding evaluation on ALOHA.**
> > >
> > > The success rate reported in our paper is **stage-wise success** defined cumulatively: success at a given stage requires successful completion of all preceding stages, i.e. the failure of the current stage will cause the failure of the following stage. This evaluation protocol has also been adopted in prior work, such as *Vision in Action: Learning Active Perception from Human Demonstrations*. Our motivation for using this protocol is that stage-wise success rates provide a clear way to demonstrate ReLAM’s ability to solve long-horizon tasks through the generation of keypoint-based subgoals and rewards.
> > >
> > > We again thank the reviewer for the thoughtful feedback. We will continuously devote to further improving the paper.

---

### Decision · Program_Chairs · 2026-04-30

**Decision:**

Accept (regular)

**Comment:**

Reviewers had somewhat mixed evaluations of this paper. However, reviewers generally found the paper well-written / well-presented. Concerns were raised about the limitations of an image keyframe-based method, as well as a potentially-questionable problem setting. Other concerns included an absence of evaluation of robustness to viewpoint change or camera motion, as well as the large amount of pending changes necessitated by new experimental results reported in author responses. Otherwise, the paper was technically sound. It may be useful to a nontrivial fraction of the ICML community.